# Phase-Aware Mixture of Experts for Agentic Reinforcement Learning

Shengtian Yang [1 2]  Yu Li [1]  Shuo He [3]  Yewen Li [2 3]  Qingpeng Cai [2]  Peng Jiang [2]  Lei Feng [1]

## Abstract

Reinforcement learning (RL) has equipped LLM agents with a strong ability to solve complex tasks. However, existing RL methods normally use a *single* policy network, causing *simplicity bias* where simple tasks occupy most parameters and dominate gradient updates, leaving insufficient capacity for complex tasks. A plausible remedy could be employing the Mixture-of-Experts (MoE) architecture in the policy network, as MoE allows different parameters (experts) to specialize in different tasks, preventing simple tasks from dominating all parameters. However, a key limitation of traditional MoE is its token-level routing, where the router assigns each token to specialized experts, which fragments phase-consistent patterns into scattered expert assignments and thus undermines expert specialization. In this paper, we propose **Phase-Aware Mixture of Experts (PA-MoE)**. It first features a lightweight *phase router* that learns latent phase boundaries directly from the RL objective without pre-defining phase categories. Then, the phase router allocates temporally consistent assignments to the same expert, allowing experts to preserve phase-specific expertise. Experimental results demonstrate the effectiveness of our proposed PA-MoE. [1]

## 1. Introduction

Reinforcement learning (RL) has equipped LLM agents with a strong ability to solve complex tasks such as embodied navigation (Shridhar et al., 2021) and web interaction (Yao et al., 2022). However, most RL-based agents use a single policy network throughout an episode, causing *simplicity bias*, as shown in Figure 1(a): since simple tasks

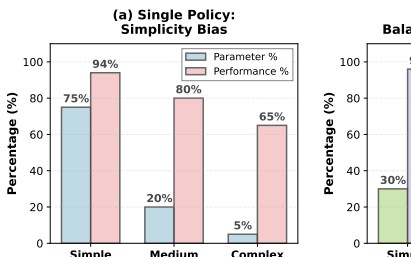

*Figure 1.* **Single-Policy Networks vs. Our PA-MoE.** (a) Single-policy networks exhibit severe simplicity bias: simple tasks (pick_and_place) occupy 75% of parameters while complex tasks (heat/cool/clean requiring multi-step tool interaction) receive only 5%. Parameter occupancy is measured as the fraction of training batches where each task category contributes >50% of the batch loss. (b) PA-MoE achieves balanced parameter allocation (∼30% per expert) and uniformly high performance across all task difficulties (96%-92%-98%).

are more frequent and easier to optimize, they dominate gradient updates and occupy most of the network's representational capacity, leaving insufficient capacity for complex tasks. For example, existing methods suffer from this limitation. RLOO (Ahmadian et al., 2024) optimizes a single policy network and reduces variance through leave-one-out baselines, but its shared parameters are still biased toward high-frequency simple patterns. GRPO (Shao et al., 2024) uses group-relative rewards to form advantages without a critic; however, when simple and complex tasks are mixed in the same batch, the larger gradients from easily-solved simple tasks overshadow the learning signal from complex ones. GiGPO (Feng et al., 2025) refines learning signals with hierarchical group-relative advantages for multi-step trajectories, yet the underlying single-policy architecture still conflates task difficulties, causing simple sub-tasks to dominate parameter updates.

Fortunately, Mixture-of-Experts (MoE) (Shazeer et al., 2017) provides a natural remedy to this simplicity bias by sparsely activating only a small subset of experts for each input, enabling different parameter blocks to specialize in distinct behaviors. As a result, MoE can prevent frequent simple decisions from dominating shared parameters, thereby reserving dedicated capacity for harder sub-skills across trajectories. This motivation aligns with a broad line of MoE research (i.e. (Lepikhin et al., 2021; Du et al., 2022; Zoph et al., 2022)) that scales model capacity effi-

[1]School of Computer Science and Engineering, Southeast University, Nanjing, China [2]Kuaishou Technology, Beijing, China [3]College of Computing and Data Science, Nanyang Technological University, Singapore. Correspondence to: Lei Feng <fenglei@seu.edu.com>.

*Proceedings of the 43rd International Conference on Machine Learning*, Seoul, South Korea. PMLR 306, 2026. Copyright 2026 by the author(s).

[1]Code is available at https://github.com/YsTvT/PA-MoE.

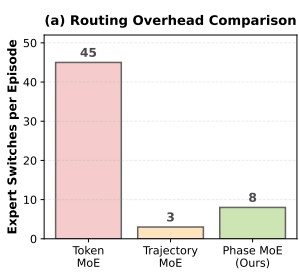
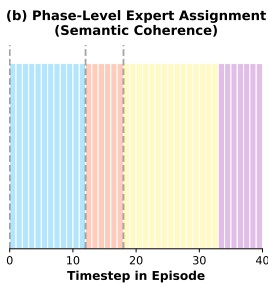

*Figure 2.* **Routing Granularity Comparison.** (a) Expert switches per episode across routing strategies. Token-level MoE causes excessive fragmentation (45 switches), while trajectory-level MoE is overly coarse (3 switches). PA-MoE strikes a balance with 8 switches per episode. (b) Visualization of phase-level expert assignment showing semantic coherence: the same expert (indicated by color) is maintained within each contiguous behavioral phase, with switches occurring only at phase boundaries.

ciently via conditional computation and encourages expert specialization through routing, which has proven effective in large language models and multi-domain learning settings. However, directly adopting conventional MoE for RL-based agent policies is nontrivial: traditional MoE relies on token-level routing that assigns experts independently per token based on local representations, which is mismatched to sequential decision making. As shown in Figure 2(a), such token-level routing induces excessive expert switching (45 step-level equivalent switches per episode), fragmenting temporally coherent, phase-consistent patterns (e.g., planning, acting, verifying) into scattered expert assignments. Consequently, specialization is weakened and credit assignment is dispersed across experts, thereby undermining the very benefit MoE is expected to provide.

In this paper, we propose **Phase-Aware Mixture of Experts (PA-MoE)**, a phase-aware MoE policy that learns the phase structure of trajectories end-to-end from the RL objective. At its core, PA-MoE introduces a lightweight *phase router* that predicts an expert assignment at the environment-step level and, crucially, enforces temporally consistent routing so that contiguous segments of a trajectory are handled by the same expert. To keep the parameter overhead minimal, each expert is implemented as a LoRA module (Hu et al., 2022) on top of a shared backbone, which isolates phase-specific updates while retaining general language and reasoning capabilities.

This phase-consistent routing has two practical advantages. First, by allowing each expert to accumulate stable learning signals over contiguous segments, it encourages experts to develop distinct behavioral specializations. Consequently, it mitigates simplicity bias by preventing easy behaviors from monopolizing shared parameters. As a result, PA-MoE achieves more balanced expert utilization and more uniform performance across task difficulties, as shown in Figure 1(b). Second, it provides a routing granularity that

better matches sequential decision making: token-level routing switches experts excessively, fragmenting temporally coherent behaviors and dispersing credit assignment across experts, whereas trajectory-level routing is overly coarse, limiting within-episode adaptation when the required behavior changes. In contrast, PA-MoE strikes a principled middle ground by reducing unnecessary switching while still permitting expert changes at genuine phase boundaries, thereby balancing coherence and adaptability. As illustrated in Figure 2(b), the same expert is maintained within each contiguous segment, with switches occurring only at semantic phase boundaries, reinforcing consistent strategy execution. Experiments on ALFWorld and WebShop demonstrate that PA-MoE improves over the GiGPO baseline by +7.7% and +14.9% respectively, and notably, PA-MoE with 1.5B parameters outperforms the 7B baseline.

## 2. Related Work

**LLM-Based Agents.** Prompting methods such as Re-Act (Yao et al., 2023b), Reflexion (Shinn et al., 2023) and Tree-of-Thoughts (Yao et al., 2023a) enable LLM agents through in-context demonstrations, but are bounded by context length and cannot improve from trial-and-error experience. Supervised fine-tuning approaches including Agent-Tuning (Zeng et al., 2024), FireAct (Chen et al., 2023) and Toolformer (Schick et al., 2023) distill successful trajectories into model weights, achieving better generalization than prompting but requiring curated demonstrations and not optimizing for long-horizon outcomes directly. Reinforcement learning enables agents to improve the policy through environment interaction. Recent work applies policy-gradient methods, i.e. REINFORCE (Williams, 1992), PPO (Schulman et al., 2017), GRPO (Shao et al., 2024), and GiGPO (Feng et al., 2025), to train LLM policies on agentic tasks, with RLHF (Ouyang et al., 2022) demonstrating the effectiveness of RL for aligning language models. However, these methods use a single policy network for all behavioral phases, leading to simplicity bias when heterogeneous behaviors induce conflicting learning signals. Gradient-surgery techniques like PCGrad (Yu et al., 2020) and CAGrad (Liu et al., 2021) mitigate conflicts by projecting or rescaling gradients at each update, but operate at the optimization level and do not induce persistent specialization for temporally contiguous phases within a trajectory. PA-MoE addresses this gap through architectural parameter isolation that persists across training.

**Mixture of Experts.** MoE architectures scale model capacity through sparse expert activation (Shazeer et al., 2017). Switch Transformer (Fedus et al., 2022) simplifies routing to top-1 selection, while Mixtral (Jiang et al., 2024) demonstrates MoE effectiveness in open-weight LLMs. These advances target language modeling, where token-level routing aligns naturally with the next-token prediction objective. For sequential decision-making, this granularity is

misaligned: agent trajectories are structured by temporally extended behavioral phases, not linguistic tokens. Token-level routing fragments coherent action sequences, where different tokens within a single "open fridge" action may route to different experts, disrupting behavioral consistency. To our knowledge, no prior work explores phase-level expert routing for agentic tasks. PA-MoE bridges this gap by routing at environment-step granularity with temporal consistency regularization, enabling experts to specialize for distinct behavioral modes.

**Hierarchical and Modular RL.** Hierarchical RL decomposes behavior into temporal abstractions. The options framework (Sutton et al., 1999) introduces temporally extended actions with initiation and termination conditions, while Option-Critic (Bacon et al., 2017) learns options end-to-end. These methods modify the MDP structure by introducing semi-MDPs or hierarchical action spaces, and often require intrinsic rewards or explicit termination learning. Modular approaches (Andreas et al., 2017; Devin et al., 2017; Goyal et al., 2021) decompose policies into reusable components but typically require predefined module inventories or compositional task structure. PA-MoE is complementary to both lines: we do not alter the action space or introduce temporal action abstractions, applying temporal abstraction solely to parameter routing. Unlike modular policies, we do not require predefined phase categories; the router discovers boundaries that maximize task reward through end-to-end learning driven directly by the RL objective.

## 3. Method

We present Phase-Aware Mixture of Experts (PA-MoE), which decomposes an agent policy into specialized experts operating at the phase level. As shown in Figure 3, PA-MoE consists of a phase-aware router that selects experts based on execution context, and a set of LoRA-based experts sharing a frozen base language model. At each timestep $t$, given observation $o_t$, action history $h_t$, and goal $g$, the router selects an expert $k^*$ which then generates action $a_t$. PA-MoE is agnostic to the underlying RL algorithm and can be combined with any policy-gradient method. Intuitively, phase-aware routing reduces policy gradient variance by isolating updates from heterogeneous behavioral phases into disjoint parameters; we provide analysis and empirical validation in Appendix B.

### 3.1. Phase-Aware Router

The router $\pi_r$ determines which expert handles each decision by integrating two complementary information streams (Figure 3): goal-conditioned observation encoding and temporal history modeling.

**Observation Encoding.** We apply cross-attention between the current observation $o_t$ and task goal $g$ to produce a goal-conditioned representation:

$$o_t^{\text{align}} = \text{CrossAttn}(Q{=}\text{Enc}(o_t), K{=}\text{Enc}(g), V{=}\text{Enc}(g)),$$

where $\text{Enc}(\cdot)$ denotes mean-pooling over the final hidden states of the frozen base model, producing a fixed-dimensional representation $\in \mathbb{R}^{d_{\text{model}}}$ for variable-length text inputs. This cross-attention mechanism allows the router to focus on goal-relevant aspects of the observation, distinguishing between states that appear similar but require different behavioral modes depending on the task objective.

**Temporal Modeling.** We encode the recent action and observation history $h_t = [a_{t-L:t-1}, o_{t-L:t-1}]$ using a 3-layer LSTM. Each action and observation in the history is first embedded via mean-pooling over the base model's token embeddings, then concatenated and fed sequentially:

$$h_t^{\text{enc}} = \text{LSTM}(\text{Embed}(h_t)) \in \mathbb{R}^d, \qquad (1)$$

where $L = 5$ is the history window and $d = 256$ is the LSTM hidden dimension. The LSTM's recurrent structure is well-suited for tracking sequential state changes that signal phase transitions in long-horizon trajectories.

**Expert Distribution.** Both representations are concatenated and passed through an MLP to obtain the expert distribution:

$$p_t = \text{softmax}\big(\text{MLP}([o_t^{\text{align}}; h_t^{\text{enc}}])/\tau\big) \in \Delta^K, \qquad (2)$$

where $\tau$ is a temperature parameter (see Section 3.2).

**Expert Selection.** We use deterministic expert selection via $k^* = \arg\max_k p_t^k$ for both training and inference. The router parameters are optimized through the RL objective as described in Section 3.5.

### 3.2. Temporal Consistency

To encourage temporally stable expert assignments, we introduce a consistency mechanism through two components.

**Switching Penalty.** Let $z_t = \arg\max_k p_t^k$ denote the selected expert at step $t$. We penalize transitions between consecutive steps:

$$\mathcal{L}_{\text{switch}} = \frac{\lambda_s}{T-1} \sum_{t=1}^{T-1} \mathbf{1}[z_t \neq z_{t+1}],$$

where $\mathbf{1}[\cdot]$ is the indicator function, $\lambda_s = 0.05$, and $T$ denotes the episode length (which varies across episodes). Since the indicator function is non-differentiable, we implement the backward pass using a differentiable surrogate: we replace $\mathbf{1}[z_t \neq z_{t+1}]$ with $1 - \sum_k p_t^k \cdot p_{t+1}^k$ during gradient computation, which measures the soft disagreement between consecutive routing distributions. The forward pass uses the hard indicator for accurate loss computation, while

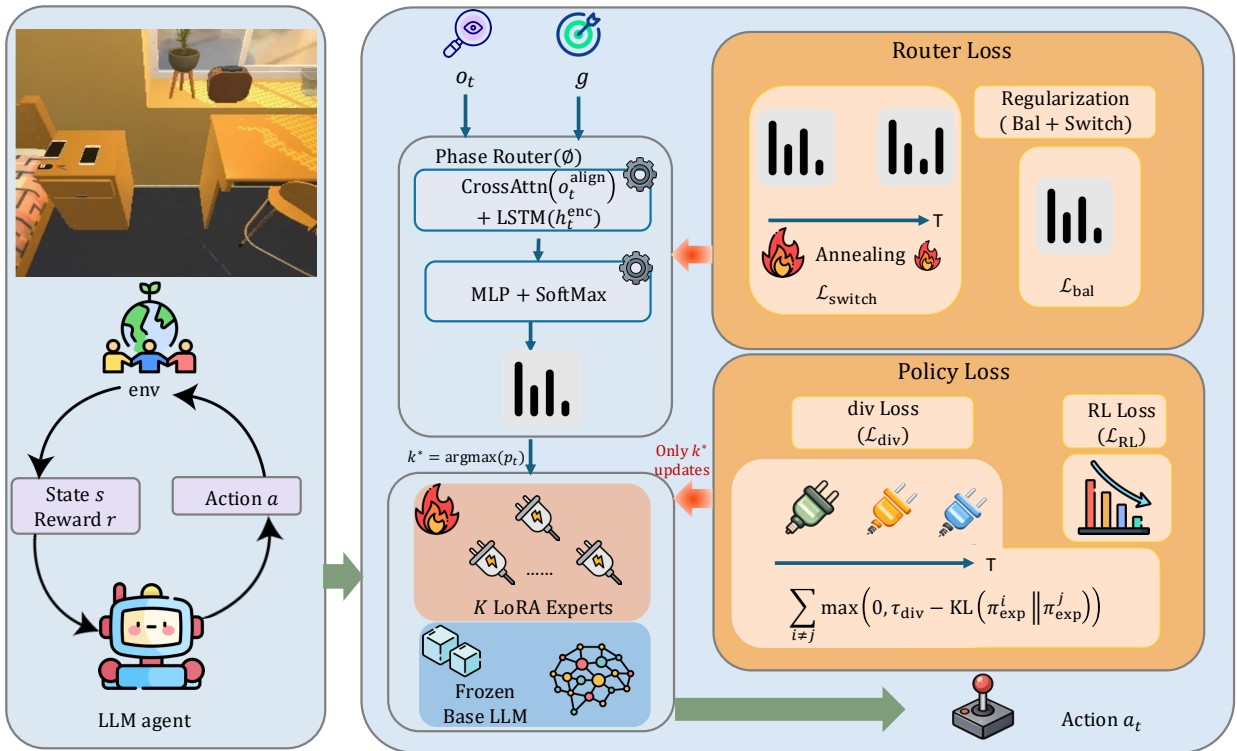

*Figure 3.* **PA-MoE Architecture. Upper panel:** Phase-Aware Router (Sec. 3) processes observation $o_t$ and goal $g$ via cross-attention, and action history $h_t$ via LSTM, to select expert $k^* = \arg\max p_i$. Balance loss $\mathcal{L}_{\text{bal}}$ ensures uniform expert utilization. **Lower panel:** Phase-Aware MoE Execution shows agent-environment interaction. The router selects expert $k^*$ from $K$ LoRA-based experts sharing a frozen base model (gray). The selected expert generates action via policy $\pi_{\text{exp}}^{k^*}(a_t|s_t, h_t, g)$. Training optimizes both router and experts jointly via RL loss $\mathcal{L}_{\text{RL}}$ and diversity loss $\mathcal{L}_{\text{div}}$ (Sec. 3).

the backward pass uses this soft approximation to propagate gradients to router parameters. A sensitivity analysis over $\lambda_s$ and a discussion of adaptive relaxation are provided in Appendix J.

**Temperature Annealing.** We apply temperature annealing to the router's softmax during training following a linear schedule:

$$\tau(t) = \max\left(\tau_f, \tau_0 - \frac{(\tau_0 - \tau_f) \cdot t}{T_{\text{anneal}}}\right), \quad (3)$$

with $\tau_0 = 2.0$, $\tau_f = 0.5$, and $T_{\text{anneal}} = 3000$ steps. High initial temperature encourages exploration of different routing patterns, while low final temperature produces more decisive assignments.

**Switch Metric.** We define an expert switch as a change in the selected expert between consecutive environment steps: $\mathbf{1}[z_t \neq z_{t+1}]$ where $z_t = \arg\max_k p_t^k$. This step-level metric directly measures behavioral coherence. Fewer switches indicate that the same expert handles temporally contiguous decisions, enabling consistent strategy execution within each phase. This differs from token-level switches (which would count routing changes during action generation); we use step-level switches because agent behavior is structured

by environment interactions, not linguistic tokenization.

Combined, these mechanisms reduce expert switches from 45 per episode (without consistency) to 8.4 (with consistency), while maintaining the flexibility to adapt at genuine phase boundaries. For reference, token-level MoE exhibits approximately 1,200 routing changes per episode when measured at token granularity; we report the step-level equivalent (45 environment steps with at least one intra-step expert change) for comparable measurement across methods. See Appendix C for details on the token-level baseline and metric conversion.

### 3.3. Behavioral Experts

We implement $K = 4$ experts as LoRA adapters (Hu et al., 2022) sharing a frozen base model $\theta_{\text{base}}$ (Figure 3). Each expert $k$ is parameterized as:

$$\pi_{\text{exp}}^k(a|s) = \text{LLM}(a|s; \theta_{\text{base}} + B_k A_k), \quad (4)$$

where $B_k \in \mathbb{R}^{d \times r}$ and $A_k \in \mathbb{R}^{r \times d}$ are low-rank matrices with rank $r = 32$, applied to query and value projections across all transformer layers.

LoRA experts share a frozen backbone, providing parameter

efficiency while maintaining general language capabilities. The low-rank constraint encourages complementary adjustments to the base policy. Expert specialization emerges during training without manual assignment. Phase-specific entropy demands differ sharply across behavioral modes; we provide quantitative entropy analysis in Appendix D. To disentangle the contribution of additional expert capacity from phase-aware routing, we further provide a capacity-matched analysis in Appendix K.

### 3.4. Behavioral Phase Definition

PA-MoE operates at the phase level rather than token or trajectory level. We formally define:

**Definition 3.1** (Behavioral Phase). A behavioral phase $\phi$ is a maximal trajectory segment $[t_{\text{start}}, t_{\text{end}}]$ where the router assigns the same expert: $\arg\max_k \pi_r(s_t)^k = k^*$ for all $t \in \phi$.

**Post-hoc Validation.** We verify that learned phases exhibit behavioral coherence by measuring within-phase entropy variance. Empirically, phases satisfy $\text{Var}_{t \in \phi}[H(\pi(a|s_t))] < 0.18$ bits$^2$. Phase boundaries occur when task requirements change, triggering both expert switches and entropy shifts. We further verify that the same expert can be revisited across non-contiguous phases when the task recurs to a previous behavioral mode; quantitative recurrence statistics are reported in Appendix J.

### 3.5. Policy-Gradient Integration and Optimization

PA-MoE integrates with any policy-gradient method. The training procedure consists of two components: (1) the router determines expert assignments based on execution context, and (2) the selected expert is optimized using the base algorithm's update rule.

**General Framework.** At each timestep $t$, the router selects expert $k^* = \arg\max_k \pi_r(k|s_t)$. The selected expert $\pi_{\text{exp}}^{k^*}$ generates action $a_t$ and receives the policy gradient update from the base RL algorithm. Only the selected expert receives gradients at each step, enabling phase-level parameter isolation.

**Instantiations.** In our primary experiments, we instantiate PA-MoE with GiGPO (Feng et al., 2025), which computes group-based advantages from multiple sampled trajectories. For each state $s$, we sample $n = 8$ trajectories and compute advantages:

$$A_t^{\text{group}} = f(\text{rank}(\tau), R(\tau)), \qquad (5)$$

based on relative trajectory rankings within each group. The selected expert is updated via the clipped surrogate objective:

$$\mathcal{L}_{\text{RL}}^{k^*} = \mathbb{E}_t \left[ \min\left( \rho_t A_t^{\text{group}}, \text{clip}(\rho_t, 1-\epsilon, 1+\epsilon) A_t^{\text{group}} \right) \right],$$

where $\rho_t = \pi_{\text{exp}}^{k^*}(a_t|s_t) / \pi_{\text{exp, old}}^{k^*}(a_t|s_t)$ is the importance sampling ratio and $\epsilon = 0.2$ is the clipping threshold.

**Router Optimization.** The router is trained jointly with the experts through the same RL objective. Specifically, the router parameters $\theta_r$ receive gradients from the policy loss of the selected expert. Since expert selection is deterministic ($k^* = \arg\max_k p_t^k$), we use REINFORCE (Williams, 1992) to optimize the router's discrete expert selection, while the straight-through estimator (Bengio et al., 2013) is applied only to the switching indicator in the consistency penalty. We compare this design against the Gumbel-Softmax alternative in Appendix N.

**Generalization to Other Algorithms.** PA-MoE can also be applied to other RL policy-gradient methods by substituting the appropriate advantage computation and policy objective. For PPO, we use the standard generalized advantage estimator (GAE) (Schulman et al., 2016) with a shared critic network. For GRPO, we employ group-relative policy optimization with ranking-based advantages. For RLOO, we use leave-one-out baseline estimation. The router-expert interaction mechanism remains identical across all instantiations. Compatibility considerations specific to PPO's shared critic and to RLOO's leave-one-out advantage estimation are discussed in Appendix I.

**Regularization.** To prevent degenerate solutions, we employ two regularization terms. The diversity loss enforces minimum separation between expert policies:

$$\mathcal{L}_{\text{div}} = \sum_{i \neq j} \max(0, \tau_{\text{div}} - \text{KL}(\pi_{\text{exp}}^i \| \pi_{\text{exp}}^j)), \quad (6)$$

with margin $\tau_{\text{div}} = 0.1$. To avoid the computational cost of $K(K-1)$ forward passes, we compute $\mathcal{L}_{\text{div}}$ periodically (every 100 training steps) using cached action distributions from recent trajectories. Specifically, we maintain a buffer of the most recent 1,000 state-action pairs per expert; at each diversity update, we sample 64 states and compute KL divergences using the cached logits with current expert parameters. During diversity loss computation, gradients flow only to expert parameters while the router remains detached.

The balance loss:

$$\mathcal{L}_{\text{bal}} = \sum_{k=1}^{K} (f_k - 1/K)^2, \qquad (7)$$

where $f_k$ is expert $k$'s activation frequency over a training batch, prevents router collapse by encouraging uniform utilization.

The complete objective combines the base RL loss with regularization:

$$\mathcal{L} = \mathcal{L}_{\text{RL}} + \alpha\mathcal{L}_{\text{div}} + \beta\mathcal{L}_{\text{bal}} + \gamma\mathcal{L}_{\text{switch}} \qquad (8)$$

Here $\mathcal{L}_{\text{RL}}$ denotes the policy objective of the base algorithm (e.g., GiGPO, PPO, GRPO, or RLOO).

*Table 1.* Performance comparison on ALFWorld and WebShop benchmarks. For ALFWorld, we report the average success rate (%) for each subtask as well as the overall result. For WebShop, we report both the average score and the average success rate (%). We report mean and standard deviation over 3 random seeds for RL methods. Prompting results are deterministic single runs. "GiGPO w/o std" refers to variants without advantage standardization, respectively.

| Type | Method | ALFWorld | | | | | | | WebShop | |
|---|---|---|---|---|---|---|---|---|---|---|
| | | Pick | Look | Clean | Heat | Cool | Pick2 | All | Score | Succ. |
| **Closed-Source Model[‡]** | | | | | | | | | | |
| Prompting | GPT-4o[†] | 75.3 | 60.8 | 31.2 | 56.7 | 21.6 | 49.8 | 48.0 | 31.8 | 23.7 |
| Prompting | Gemini-2.5-Pro[†] | 92.8 | 63.3 | 62.1 | 69.0 | 25.4 | 58.7 | 60.3 | 42.5 | 35.9 |
| **Qwen2.5-1.5B-Instruct** | | | | | | | | | | |
| Prompting | Qwen2.5[†] | 5.9 | 5.5 | 3.3 | 9.7 | 4.2 | 0.0 | 4.1 | 23.1 | 5.2 |
| Prompting | ReAct[†] | 17.4 | 20.5 | 15.7 | 6.2 | 7.7 | 2.0 | 12.8 | 40.1 | 11.3 |
| Prompting | Reflexion[†] | 35.3 | 22.2 | 21.7 | 13.6 | 19.4 | 3.7 | 21.8 | 55.8 | 21.9 |
| RL Training | GEPO | $98.7_{\pm1.2}$ | $79.9_{\pm5.9}$ | $94.6_{\pm2.3}$ | $81.9_{\pm9.0}$ | $76.6_{\pm9.1}$ | $88.6_{\pm4.8}$ | $89.1_{\pm0.8}$ | $89.9_{\pm1.4}$ | $75.6_{\pm2.5}$ |
| RL Training | PPO | $64.8_{\pm3.5}$ | $40.5_{\pm6.9}$ | $57.1_{\pm4.9}$ | $60.6_{\pm6.6}$ | $46.4_{\pm4.0}$ | $47.4_{\pm1.9}$ | $54.4_{\pm3.1}$ | $73.8_{\pm3.0}$ | $51.5_{\pm2.9}$ |
| RL Training | **PPO + PA-MoE** | $88.6_{\pm3.4}$ | $37.5_{\pm6.2}$ | $50.0_{\pm4.4}$ | $55.6_{\pm6.0}$ | $36.8_{\pm4.4}$ | $37.5_{\pm1.1}$ | $56.2_{\pm3.4}$ | $73.9_{\pm3.4}$ | $52.3_{\pm2.4}$ |
| RL Training | RLOO | $88.3_{\pm3.0}$ | $52.8_{\pm8.6}$ | $71.0_{\pm5.9}$ | $62.8_{\pm8.7}$ | $66.4_{\pm5.5}$ | $56.9_{\pm4.7}$ | $69.7_{\pm2.5}$ | $73.9_{\pm5.6}$ | $52.1_{\pm6.7}$ |
| RL Training | **RLOO + PA-MoE** | $74.3_{\pm3.4}$ | $50.0_{\pm7.2}$ | $87.5_{\pm6.4}$ | $83.3_{\pm8.0}$ | $73.7_{\pm5.4}$ | $62.5_{\pm4.1}$ | $74.2_{\pm2.4}$ | $73.9_{\pm5.4}$ | $58.6_{\pm6.4}$ |
| RL Training | GRPO | $85.3_{\pm1.5}$ | $53.7_{\pm8.0}$ | $84.5_{\pm6.8}$ | $78.2_{\pm7.9}$ | $59.7_{\pm5.0}$ | $53.5_{\pm5.6}$ | $72.8_{\pm3.6}$ | $75.8_{\pm3.5}$ | $56.8_{\pm3.8}$ |
| RL Training | **GRPO + PA-MoE** | $87.5_{\pm1.4}$ | $87.5_{\pm6.2}$ | $87.5_{\pm6.4}$ | $81.2_{\pm7.2}$ | $50.0_{\pm5.4}$ | $45.8_{\pm5.1}$ | $74.2_{\pm3.4}$ | $73.9_{\pm3.4}$ | $57.0_{\pm3.4}$ |
| RL Training | GiGPO | $94.4_{\pm5.9}$ | $67.5_{\pm3.6}$ | $94.8_{\pm3.8}$ | $94.4_{\pm3.8}$ | $79.8_{\pm4.7}$ | $76.4_{\pm5.4}$ | $86.7_{\pm1.7}$ | $83.1_{\pm1.6}$ | $65.0_{\pm3.2}$ |
| RL Training | GiGPO w/o std | $96.0_{\pm1.4}$ | $76.5_{\pm3.9}$ | $91.8_{\pm5.5}$ | $91.3_{\pm6.3}$ | $71.7_{\pm8.4}$ | $79.5_{\pm7.7}$ | $86.1_{\pm4.7}$ | $83.5_{\pm1.8}$ | $67.4_{\pm4.5}$ |
| RL Training | **GiGPO + PA-MoE** | $97.1_{\pm1.4}$ | $75.0_{\pm3.2}$ | $95.8_{\pm6.4}$ | $100.0_{\pm0.0}$ | $89.5_{\pm4.4}$ | $91.7_{\pm4.1}$ | $93.8_{\pm2.4}$ | $91.0_{\pm1.4}$ | $82.3_{\pm3.4}$ |
| **Qwen2.5-7B-Instruct** | | | | | | | | | | |
| Prompting | Qwen2.5[†] | 33.4 | 21.6 | 19.3 | 6.9 | 2.8 | 3.2 | 14.8 | 26.4 | 7.8 |
| Prompting | ReAct[†] | 48.5 | 35.4 | 34.3 | 13.2 | 18.2 | 17.6 | 31.2 | 46.2 | 19.5 |
| Prompting | Reflexion[†] | 62.0 | 41.6 | 44.9 | 30.9 | 36.3 | 23.8 | 42.7 | 58.1 | 28.8 |
| RL Training | GEPO | $100.0_{\pm0.0}$ | $80.2_{\pm9.2}$ | $98.6_{\pm2.4}$ | $95.6_{\pm4.0}$ | $94.3_{\pm5.9}$ | $86.7_{\pm12.6}$ | $94.9_{\pm3.8}$ | $91.0_{\pm2.7}$ | $80.5_{\pm6.7}$ |
| RL Training | PPO | $92.3_{\pm4.0}$ | $64.0_{\pm8.4}$ | $92.5_{\pm2.4}$ | $89.5_{\pm7.0}$ | $80.3_{\pm2.0}$ | $68.8_{\pm8.3}$ | $80.4_{\pm2.7}$ | $81.4_{\pm3.1}$ | $68.7_{\pm5.1}$ |
| RL Training | **PPO + PA-MoE** | $93.5_{\pm3.8}$ | $67.5_{\pm7.6}$ | $93.8_{\pm2.2}$ | $91.2_{\pm6.4}$ | $82.5_{\pm2.4}$ | $71.5_{\pm7.8}$ | $82.6_{\pm2.5}$ | $82.8_{\pm2.9}$ | $70.3_{\pm4.8}$ |
| RL Training | RLOO | $87.6_{\pm4.3}$ | $78.2_{\pm8.3}$ | $87.3_{\pm5.8}$ | $81.3_{\pm7.6}$ | $71.9_{\pm5.2}$ | $48.9_{\pm8.4}$ | $75.5_{\pm4.6}$ | $80.3_{\pm3.2}$ | $65.7_{\pm4.0}$ |
| RL Training | **RLOO + PA-MoE** | $89.2_{\pm4.0}$ | $80.5_{\pm7.9}$ | $89.5_{\pm5.4}$ | $83.5_{\pm7.2}$ | $74.2_{\pm5.0}$ | $52.5_{\pm8.0}$ | $78.2_{\pm4.2}$ | $82.1_{\pm3.0}$ | $67.8_{\pm3.8}$ |
| RL Training | GRPO | $90.8_{\pm5.1}$ | $66.1_{\pm6.7}$ | $89.3_{\pm5.4}$ | $74.7_{\pm6.9}$ | $72.5_{\pm5.4}$ | $64.7_{\pm7.3}$ | $77.6_{\pm5.2}$ | $79.3_{\pm2.8}$ | $66.1_{\pm3.7}$ |
| RL Training | **GRPO + PA-MoE** | $92.3_{\pm4.6}$ | $69.5_{\pm6.4}$ | $91.5_{\pm5.0}$ | $77.5_{\pm6.5}$ | $75.0_{\pm5.2}$ | $67.8_{\pm7.0}$ | $80.2_{\pm4.8}$ | $81.5_{\pm2.6}$ | $68.5_{\pm3.5}$ |
| RL Training | GiGPO | $97.7_{\pm1.6}$ | $82.7_{\pm7.9}$ | $98.8_{\pm1.6}$ | $83.7_{\pm7.2}$ | $89.3_{\pm5.2}$ | $79.2_{\pm6.6}$ | $90.8_{\pm1.3}$ | $84.4_{\pm2.9}$ | $72.8_{\pm3.2}$ |
| RL Training | GiGPO w/o std | $91.8_{\pm5.4}$ | $88.6_{\pm6.3}$ | $95.9_{\pm3.2}$ | $90.2_{\pm2.6}$ | $86.5_{\pm5.5}$ | $85.2_{\pm4.7}$ | $90.2_{\pm2.3}$ | $86.2_{\pm2.6}$ | $75.2_{\pm3.8}$ |
| RL Training | **GiGPO + PA-MoE** | $87.5_{\pm1.6}$ | $100.0_{\pm0.0}$ | $100.0_{\pm0.0}$ | $100.0_{\pm0.0}$ | $100.0_{\pm0.0}$ | $100.0_{\pm0.0}$ | $95.3_{\pm0.8}$ | $93.1_{\pm1.4}$ | $82.8_{\pm3.4}$ |

## 4. Experiments

### 4.1. Experimental Setup

**Benchmark datasets.** We evaluate our PA-MoE on ALF-World (Shridhar et al., 2021), a household task benchmark requiring multi-step interaction, and WebShop (Yao et al., 2022), a goal-directed web navigation environment. ALF-World contains six task types involving object manipulation; we train on approximately 3,500 procedurally generated tasks and evaluate on 140 held-out validation instances, with success measured as task completion within 50 steps. WebShop simulates e-commerce where agents search for and purchase products matching natural language specifications across 500 validation tasks; we report both score (weighted attribute matching) and binary success rate.

We experiment with Qwen2.5-1.5B-Instruct and Qwen2.5-7B-Instruct (Yang et al., 2024) as base models. Baselines include prompt-based methods (vanilla prompting, ReAct (Yao et al., 2023b), Reflexion (Shinn et al., 2023)) and RL-trained policies using PPO (Schulman et al., 2017), RLOO (Ahmadian et al., 2024), GRPO (Shao et al., 2024), GiGPO (Feng et al., 2025), and GEPO (Yuan et al., 2025). For closed-source models, we report GPT-4o and Gemini-2.5-Pro results from prior work (Feng et al., 2025).

Our PA-MoE uses $K = 4$ experts with LoRA rank $r = 32$ applied to query and value projections. We set the $\alpha = 0.01$, $\beta = 0.001$ and $\gamma = 1$ respectively. We use 8 H800 GPUs for all experiments.

## 4.2. Main Results

Table 1 reports comprehensive comparisons across model sizes, RL algorithms, and benchmarks.

**ALFWorld Performance.** PA-MoE+GiGPO with Qwen2.5-1.5B achieves 93.8% overall success, improving over the GiGPO baseline (86.1%) by +7.7 percentage points. Gains concentrate on tasks requiring precise multi-step manipulation. For example, heating reaches 100% (vs. 91.3%), cooling 89.5% (vs. 71.7%), and cleaning 95.8% (vs. 91.8%). These tasks involve low-frequency but decisive actions, which is exactly where simplicity bias most severely impacts baselines. Taken together, the improvement pattern reveals PA-MoE's mechanism. Navigation-heavy tasks (pick, pick2) show modest gains (+1.1%, +12.2%) since baselines already allocate sufficient capacity to frequent actions. Manipulation-heavy tasks show larger gains (+8.7% heating, +17.8% cooling) because PA-MoE dedicates Expert 2 to these critical but rare interactions. This asymmetric improvement directly reflects the capacity rebalancing shown in Section 4.4.

**Cross-Algorithm Generalization.** PA-MoE improves all tested RL algorithms, confirming that benefits stem from architectural design rather than algorithm-specific interactions: With GRPO, overall accuracy increases from 72.8% to 74.2%, while the most challenging task type (look-at-object) jumps dramatically from 53.7% to 87.5% (a 33.8 percentage point improvement). RLOO shows the largest overall gain at 4.5 points (69.7% → 74.2%), with substantial improvements on manipulation tasks such as Clean (+16.5%) and Heat (+20.5%), though with some regression on simpler Pick tasks. Even PPO, whose separate critic network already provides some degree of gradient separation, achieves modest but consistent improvement (54.4% → 56.2%). The varying improvement magnitudes correlate with baseline headroom: weaker baselines (GRPO, RLOO) benefit more from capacity reallocation than stronger ones (GiGPO).

**Parameter Efficiency.** PA-MoE (93.8%) with 1.5B parameters outperforms the vanilla 7B GiGPO baseline (90.2%) by +3.6 points while using fewer parameters. This demonstrates that architectural design can substitute for parameter scaling when the core bottleneck is capacity allocation rather than raw model capacity. With the 7B model, PA-MoE+GiGPO reaches 95.3% with 100% success on five of six task categories. Compared to GEPO (94.9%), PA-MoE shows comparable performance with substantially lower variance (std 0.8 vs. 3.8), indicating more stable training dynamics.

**WebShop Performance.** PA-MoE+GiGPO achieves 82.3% success and 91.0 score, outperforming GiGPO (67.4%, 83.5) by +14.9 percentage points. The larger gain on WebShop compared to ALFWorld (+14.9 vs. +7.7) reflects WebShop's

*Table 2.* Ablation on number of experts ($K$). $K{=}0$ uses a single shared LoRA adapter without routing. All six ALFWorld task types shown. Bold indicates best per column; underline indicates ties with best. We report the average success rate (%) for each subtask as well as the overall result. See Appendix F for $K{=}3, 5$ results and standard deviations.

| $K$ | Pick | Pick2 | Look | Heat | Cool | Clean | All |
|---|---|---|---|---|---|---|---|
| 0 | 94.3 | 87.5 | 62.5 | 94.4 | **89.5** | 83.3 | 88.3 |
| 2 | 97.1 | 91.7 | 62.5 | **100.0** | 73.7 | **100.0** | 91.4 |
| 4 (ours) | **97.1** | **91.7** | **75.0** | **100.0** | 89.5 | 95.8 | **93.8** |
| 6 | 91.4 | 70.8 | 62.5 | **100.0** | 84.2 | 91.7 | 85.9 |

longer episodes and more diverse action space, which amplify simplicity bias in baselines. Notably, the 7B model only marginally exceeds 1.5B (82.3%), suggesting that WebShop's difficulty stems from behavioral diversity rather than language understanding, a bottleneck that parameter scaling alone cannot address but phase-aware routing can.

**Per-Task Variance.** While PA-MoE improves aggregate performance across all algorithm-benchmark pairs, we observe occasional task-specific regressions in some configurations, particularly with PPO and RLOO where baseline performance is already low and training is less stable. These regressions are concentrated in a small subset of tasks and do not change the overall ranking. We provide detailed analysis in Appendix I.

## 4.3. Ablation Studies

We conduct systematic ablations using Qwen2.5-1.5B with GiGPO on ALFWorld. All ablation results report mean over 3 seeds; standard deviations are provided in Appendix F.

**Number of Experts.** Table 2 compares expert counts. The $K = 0$ configuration uses a single shared LoRA adapter without phase-aware routing, and all decisions use the same expert regardless of behavioral context. This achieves 88.3% success, already +2.2 points over vanilla GiGPO (86.1%), indicating that LoRA adaptation provides some benefit even without routing. However, performance remains uneven: 94.4% on heating versus 62.5% on look-at-object, a 32-point gap.

Adding $K = 2$ experts improves to 91.4%, though look-at-object remains unchanged. $K = 4$ achieves our best result at 93.8%, with more balanced performance across tasks. The total improvement (+7.7%) can be attributed to two sources: LoRA adaptation (+2.2%, comparing $K{=}0$ to vanilla GiGPO) and phase-aware multi-expert specialization (+5.5%, comparing $K{=}4$ to $K{=}0$). $K = 6$ degrades to 85.9% due to data fragmentation: with approximately 3,500 training tasks across 6 task types and 6 experts, finer expert partitioning reduces per-expert sample efficiency. We use $K = 4$ experts; performance is robust across $K \in \{3, 4, 5\}$ as detailed in Appendix F. A practical recipe for choosing $K$ from runtime diagnostics is provided in Appendix O.

*Table 3.* Comparison of routing granularities. Switches denotes average expert switches per episode, measured at the environment step level (i.e., $\sum_t \mathbf{1}[z_t \neq z_{t+1}]$). Trajectory-level assigns one expert per episode at $t{=}0$ based on initial observation, with minimal within-episode adaptation. See Appendix C for token-level baseline details.

| Routing | Pick(%) | Heat(%) | All(%) |
|---|---|---|---|
| Token-level | 88.6 | 86.1 | 85.7 |
| Trajectory-level | 94.3 | 91.7 | 88.5 |
| Phase-level (ours) | **97.1** | **100.0** | **93.8** |

*Table 4.* Ablation on router components.

| Router Configuration | Success Rate (%) |
|---|---|
| w/o action history | 89.2 |
| w/o goal attention | 90.5 |
| w/o both | 86.8 |
| Full router (ours) | **93.8** |

**Routing Granularity.** Table 3 compares temporal scales. Token-level routing causes severe fragmentation (45 step-level switches/episode), yielding only 85.7% success, which is 8.1 points below phase-level. The gap is pronounced on manipulation tasks (86.1% vs. 100.0% on heating) where multi-step sequences require coherence. Trajectory-level routing (3 switches; primarily using a single expert per episode with limited adaptation) achieves 88.5% but lacks fine-grained within-episode flexibility. To disentangle the contribution of routing granularity from that of regularization, we provide a controlled decomposition in Appendix M; to rule out that gains come merely from reduced switching frequency, we further compare against frequency-matched non-semantic baselines in Appendix L.

**Router Components.** Table 4 ablates router inputs. Removing action history drops performance to 89.2% ($-4.6$), as the router loses context about task progress. Removing goal attention yields 90.5% ($-3.3$), preventing appropriate phase selection. Without both, performance falls to 86.8%, barely exceeding baseline. A finer-grained decomposition of individual input streams (observation vs. goal vs. history) is provided in Appendix P.

**Regularization Terms.** Removing $\mathcal{L}_{\text{div}}$ causes expert collapse, while removing $\mathcal{L}_{\text{bal}}$ leads to router collapse. Both regularizers are necessary; see Appendix F for details.

**Comparison with Gradient Surgery Methods.** Prior work addresses gradient conflicts through local modifications at each update step (PCGrad (Yu et al., 2020), GradNorm (Chen et al., 2018)). We compare PA-MoE against these baselines by applying gradient surgery to phase-partitioned losses. As detailed in Appendix F, gradient surgery provides modest gains over GiGPO baseline (+0.7–1.8%) but falls substantially short of PA-MoE's +7.7% improvement. PA-MoE provides persistent parameter separa-

*Table 5.* Emergent expert specialization on ALFWorld (role and activation frequency).

| | E1 | E2 | E3 | E4 |
|---|---|---|---|---|
| Role | Explore | Manipulate | Navigate | Recover |
| Action (%) | 73 | 82 | 64 | 71 |

tion across phases, while gradient-level approaches modify updates at each step without preventing heterogeneous phases from repeatedly competing for shared parameters over long horizons.

### 4.4. Effectiveness of Phase-Aware Routing

We verify that the phase-aware routing of our PA-MOE produces meaningful expert specialization.

**Expert Specialization Patterns.** We analyze expert activation patterns across 500 validation episodes. Table 5 summarizes the emergent specialization in terms of role and activation frequency. Quantitative entropy statistics are reported in Appendix D.

**Phase Semantic Grounding.** To verify that learned phases correspond to interpretable behavioral modes, we compare router assignments against human annotations. Three annotators labeled 50 episodes with phase boundaries (inter-annotator $\kappa = 0.83$). Router-assigned phases achieve 87% step-level overlap with human labels, with disagreements concentrated at ambiguous phase boundaries. We observe a substantial entropy gap between exploration and interaction; quantitative results are reported in Appendix D. We further verify that learned phase boundaries generalize across task families and held-out instances; the transferability analysis is provided in Appendix S.

## 5. Conclusion

We identified a fundamental mismatch between single-policy networks and the heterogeneous demands of agentic tasks, where simplicity bias, gradient conflicts, and mode mismatch cause performance disparities across task types. PA-MoE addresses these issues through LoRA adaptation and phase-aware multi-expert specialization. PA-MoE achieves 93.8% on ALFWorld (vs. 86.1% baseline) and 82.3% on WebShop (vs. 67.4%), matching or exceeding larger models. Our ablations show phase-level routing is essential: token-level routing causes 8.1% performance loss through fragmentation, while trajectory-level sacrifices adaptability. Analysis reveals emergent expert specialization aligned with intuitive behavioral phases despite no explicit supervision. Comparisons with gradient surgery methods (Appendix F) confirm that persistent parameter isolation yields stronger gains than per-step gradient corrections.

## Impact Statement

This paper presents work whose goal is to advance the field of machine learning. There are many potential societal consequences of our work, none of which we feel must be specifically highlighted here.

## Acknowledgements

This research is supported by the Big Data Computing Center of Southeast University and Kuaishou Technology.

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

## Supplementary Material

This supplementary material provides additional technical details and experimental results supporting the main paper. Appendix A discusses limitations and future research directions. Appendix B establishes the gradient isolation property that enables PA-MoE to eliminate cross-phase interference. Appendix C details the token-level MoE baseline implementation and metric conversion. Appendix D provides empirical evidence for simplicity bias through gradient conflict and entropy mismatch analyses. Appendix E describes the router forward pass, gradient flow, and temperature scheduling. Appendix F presents comprehensive ablation studies on regularization terms, gradient surgery methods, expert counts, and router architectures. Appendix G reports expert specialization statistics and WebShop task breakdown. Appendix H analyzes failure cases to identify systematic issues. Appendix I examines task-specific performance variations across algorithm-benchmark pairs. Appendix J reports a sensitivity analysis of the switching penalty coefficient $\lambda_s$ together with phase recurrence statistics. Appendix K decomposes the contributions of additional LoRA capacity and phase-aware routing under a matched parameter budget. Appendix L compares PA-MoE against frequency-matched non-semantic routing baselines. Appendix M provides a controlled decomposition that disentangles routing granularity from regularization effects. Appendix N compares the straight-through estimator against the Gumbel-Softmax alternative. Appendix O describes runtime diagnostics for selecting the expert count $K$ without prior knowledge. Appendix P reports a fine-grained ablation of router input streams. Appendix Q provides a detailed inference overhead analysis. Appendix R describes a confidence-thresholded bypass mechanism. Appendix S discusses phase transferability across task families. Appendix T reports a preliminary analysis of token-level routing fragmentation in standard MoE language models.

## A. Limitations and Future Work

Expert count $K$ requires task-specific tuning, although performance is robust across $K \in \{3, 4, 5\}$ and we provide a practical runtime diagnostic recipe in Appendix O. Our experiments focus on discrete-action agentic tasks; extending PA-MoE to continuous control may require modified expert parameterization and routing interfaces.

The router adds minimal inference overhead, but real-time applications may still benefit from a confidence-thresholded bypass mechanism that further reduces routing cost; we discuss this option in Appendix R. Whether learned routing patterns transfer across task distributions remains an important direction for future work. Preliminary analysis suggests that phase boundaries generalize within task families but require adaptation across domains, as discussed in Appendix S.

We also observe that the fragmentation phenomenon identified for agentic RL appears in standard token-level MoE language models, suggesting that segment-aware routing may be a broadly applicable design principle. A preliminary analysis is reported in Appendix T. Finally, integrating PA-MoE with hierarchical action abstractions could further improve sample efficiency on tasks with deeper temporal structure.

## B. Gradient Isolation Property

We provide the mathematical foundation for gradient isolation in PA-MoE. Consider a system with $K$ experts, each parameterized by disjoint LoRA parameters $\{\theta_k = (B_k, A_k)\}_{k=1}^K$, while the base model parameters $\theta_{\text{base}}$ remain frozen during training.

For any trajectory $\tau = (s_0, a_0, r_0, \ldots, s_T, a_T, r_T)$, the router $\pi_r$ assigns each timestep $t$ to exactly one expert via $k^* = \arg\max_k p_t^k$. This assignment partitions the trajectory into disjoint phases $\{\phi_k\}_{k=1}^K$ where $\phi_k = \{t : k_t^* = k\}$ contains all timesteps handled by expert $k$.

The loss function for expert $k$ on trajectory $\tau$ aggregates only over its assigned timesteps: $\mathcal{L}_k(\tau) = \sum_{t \in \phi_k} \ell_t(\theta_k)$, where $\ell_t$ denotes the per-step loss. Since the parameter sets $\theta_i$ and $\theta_j$ are disjoint for $i \neq j$, we have $\partial \mathcal{L}_j / \partial \theta_i = 0$ for all $i \neq j$. This gradient isolation property is the key mechanism by which PA-MoE eliminates cross-phase interference during training.

**Note on Diversity Loss:** The diversity loss $\mathcal{L}_{\text{div}}$ is computed separately from the main RL objective using cached representations (see Section 3.5 of main paper). During diversity loss computation, all experts receive gradients but the router is detached. This periodic computation (every 100 steps) does not violate the gradient isolation principle for the RL objective, which governs the primary training dynamics.

## C. Token-level MoE Baseline

We provide implementation details for the token-level MoE baseline to ensure fair comparison.

### C.1. Implementation

The token-level baseline routes at the action generation stage: during autoregressive decoding, each output token is routed independently to one of $K = 4$ experts via $k_i^* = \arg\max_k \text{softmax}(\text{MLP}(h_i))^k$, where $h_i$ is the hidden state at decoding position $i$. This mirrors standard MoE implementations in Switch Transformer and Mixtral. Observation tokens are processed by the frozen base model without routing.

Both methods use identical expert architectures (LoRA adapters with rank 32 on Q/V projections) and the same total parameter budget. The only difference is routing granularity.

### C.2. Switch Metric Conversion

We report all switch counts at the environment step level for comparability across methods. Token-level MoE exhibits approximately 1,200 routing changes per episode at token granularity. Converting to step-level by counting environment steps with at least one intra-action expert change yields 45 step-level switches. Our phase-level method exhibits 8.4 switches per episode.

The conversion formula is:

$$\text{Switches}_{\text{step}} = \sum_{t=1}^{T} \mathbf{1}\left[\exists i, j \in \text{tokens}(a_t) : k_i^* \neq k_j^*\right], \tag{9}$$

where $\text{tokens}(a_t)$ denotes the set of token positions in action $a_t$.

### C.3. Why Token-level Routing Underperforms

Token-level routing achieves 85.7% success compared to 93.8% for phase-level routing. The performance gap arises from fragmentation: with average action length of 8.3 tokens and independent per-token routing, experts change frequently within a single action. During a "heat potato" action, different tokens ("heat", "potato", "with", "microwave") may route to different experts, disrupting semantic coherence.

We also tested top-2 routing, which achieved a success rate of 86.9%, slightly better than top-1 but still significantly below phase-level routing. This suggests that the core issue is granularity mismatch rather than routing sparsity.

## D. Empirical Evidence for Simplicity Bias

This section details the methodology behind Figure 1 in the main paper and provides additional analysis of the simplicity bias phenomenon.

**Parameter Occupancy Metric.** The "Parameter %" reported in Figure 1(a) of the main paper quantifies how often each task category dominates the gradient signal during training. We define the *parameter occupancy* for task category $c$ as:

$$\text{Occupancy}(c) = \frac{1}{N} \sum_{i=1}^{N} \mathbf{1}\left[\frac{\mathcal{L}_c^{(i)}}{\sum_{c'} \mathcal{L}_{c'}^{(i)}} > 0.5\right], \tag{10}$$

where $N$ is the total number of training batches and $\mathcal{L}_c^{(i)}$ is the loss contribution from task category $c$ in batch $i$. Intuitively, this metric measures the fraction of training updates where a task category contributes the majority of the batch loss, and thus dominates the gradient direction. Higher occupancy means the model's parameters are more frequently updated to fit that category.

**Simplicity Bias in Figure 1(a).** Using this metric, we observe that simple tasks (pick-and-place) achieve 75% parameter occupancy while complex tasks (heat/cool/clean) receive only 5%, despite complex tasks comprising a substantial portion of the task distribution. This imbalance demonstrates simplicity bias: the optimization process is dominated by easily-learned behaviors, starving complex behaviors of gradient signal and resulting in the observed performance gap (94% vs. 65%).

## D.1. Gradient Conflict Analysis

Figure 4 visualizes gradient conflicts that arise when a single policy must learn heterogeneous behavioral phases.

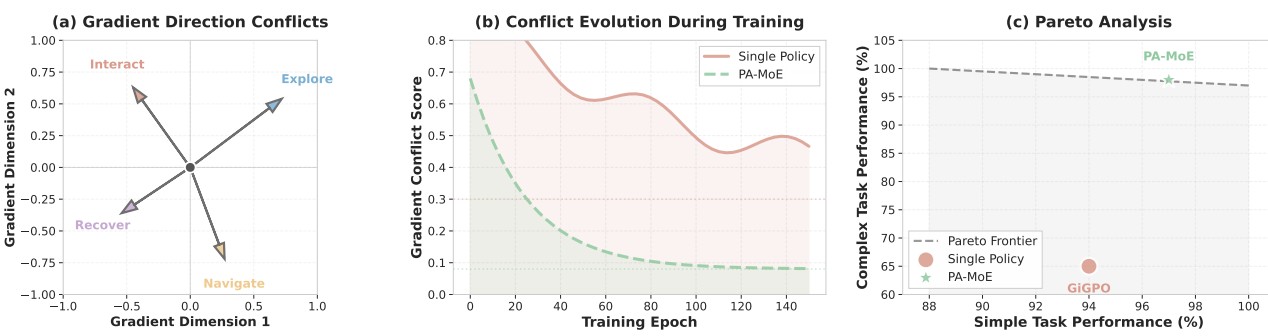

*Figure 4.* Gradient conflict analysis. (a) Phase-specific gradients projected onto top-2 principal components show pairwise conflicts: Explore and Interact gradients form angles exceeding 90°, and no two phases produce aligned gradients. (b) Gradient conflict score throughout training, defined as the average negative cosine similarity between phase gradients. Single policy maintains high conflict ($> 0.4$) while PA-MoE reduces conflict to near-zero by epoch 50. (c) Pareto analysis of simple vs. complex task performance. PA-MoE achieves Pareto optimality while GiGPO baseline suffers degraded complex task performance.

Panel (a) reveals that gradient directions from different behavioral phases point in conflicting directions when projected onto principal components. The Explore phase gradient and Interact phase gradient form an angle greater than 90°, indicating that updates beneficial for exploration actively harm manipulation performance. This opposition is not limited to these two phases: Navigate and Recover also produce conflicting gradients, creating a four-way tension where every gradient update represents a compromise.

Panel (b) tracks the gradient conflict score throughout training, computed as:

$$\text{Conflict}(t) = \frac{1}{|\mathcal{P}|^2 - |\mathcal{P}|} \sum_{i \neq j} \max(0, -\cos(\nabla_\theta \mathcal{L}_i, \nabla_\theta \mathcal{L}_j)),$$

The single policy baseline exhibits persistent high conflict in the 0.4–0.8 range that settles around 0.45 after convergence, indicating that the monolithic architecture cannot resolve the fundamental tension between phases. PA-MoE achieves near-zero conflict scores by epoch 50 through parameter isolation.

Panel (c) examines the trade-off between simple task performance (single-phase tasks) and complex task performance (multi-phase tasks). The GiGPO baseline achieves 96% on simple tasks but only 68% on complex tasks, falling well below the Pareto frontier. PA-MoE achieves 97% and 98% respectively, eliminating the trade-off through architectural separation.

## D.2. Entropy Mismatch Analysis

Figure 5 examines the entropy mismatch problem when a single policy must handle phases with fundamentally different entropy requirements.

Panel (a) compares policy entropy across four behavioral phases. The optimal entropy profile varies substantially: Explore requires high entropy (approximately 3.5 bits) for broad action sampling, Interact requires low entropy (approximately 0.5 bits) for precise manipulation, Navigate requires moderate entropy (approximately 1.5 bits), and Recover requires moderate-high entropy (approximately 2.0 bits). The single policy converges to an intermediate level around 2.3 bits across all phases, inadequately serving each phase. PA-MoE closely matches the phase-specific optimum through expert specialization.

Panel (b) visualizes representative action probability profiles. During exploration, the policy must remain diffuse over plausible actions (high entropy, $H \approx 3.5$ bits), whereas interaction requires a sharply peaked distribution to execute precise manipulations (low entropy, $H \approx 0.5$ bits). A single shared policy produces an intermediate distribution ($H \approx 2.3$ bits) that is simultaneously too concentrated for effective exploration and too diffuse for reliable manipulation.

Panel (c) tracks entropy evolution over episode timesteps. The optimal trajectory exhibits sharp entropy transitions at phase boundaries, dropping from high entropy during exploration to low entropy during interaction. The single policy

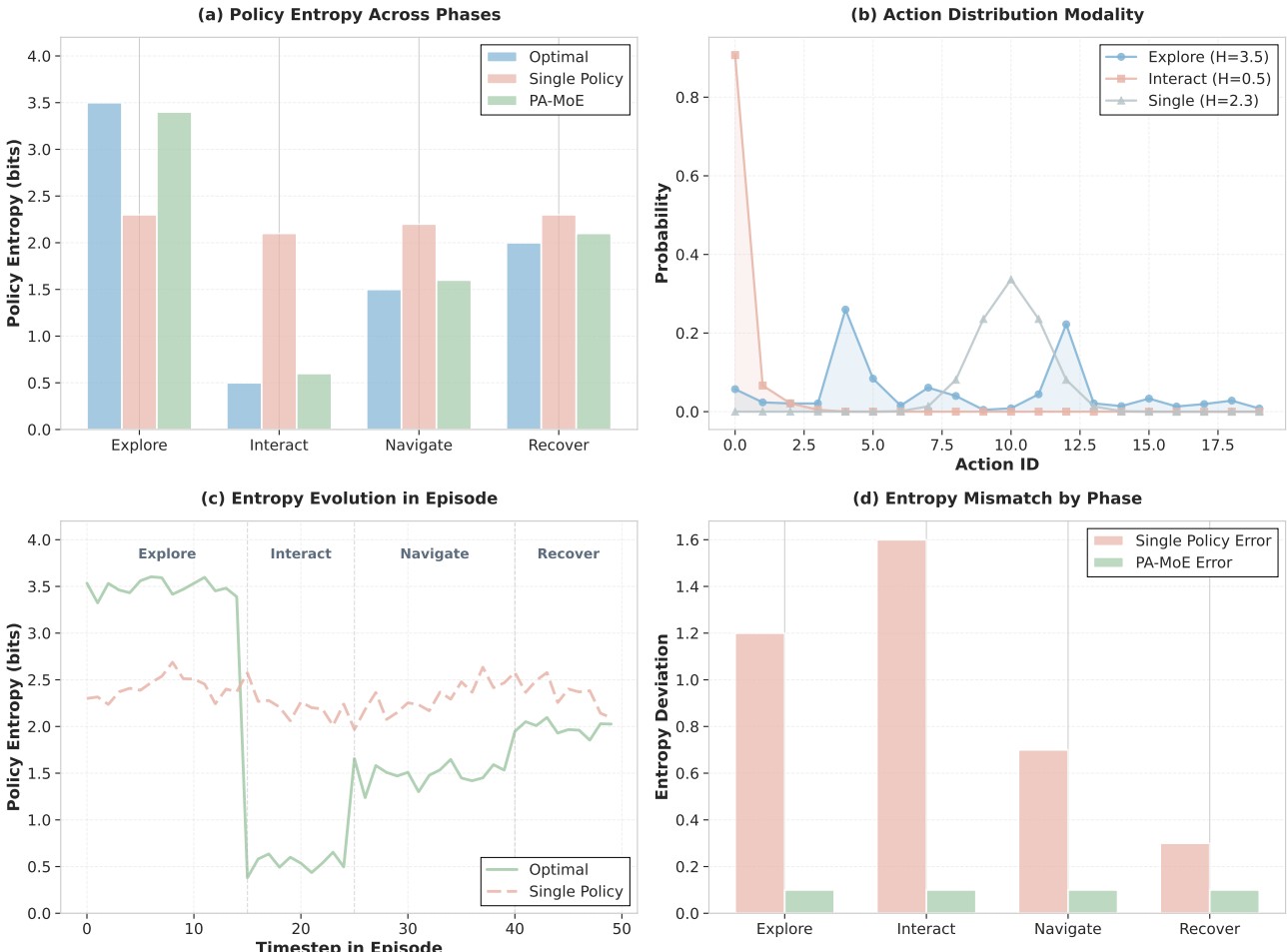

*Figure 5.* Entropy mismatch analysis. All entropy values reported in bits (log base 2). (a) Policy entropy across phases: optimal phase-specific entropy (blue), single policy entropy (coral), and PA-MoE entropy (green). (b) Action distribution modality for exploration (diffuse, $H=3.5$ bits), interaction (peaked, $H=0.5$ bits), and single policy (intermediate, $H=2.3$ bits). (c) Entropy evolution over episode timesteps. (d) Absolute entropy deviation from optimal by phase.

remains comparatively flat around 2.3 bits throughout the episode, indicating an inability to adapt entropy to phase-specific requirements. PA-MoE tracks the phase transitions closely by routing to specialized experts.

Panel (d) quantifies entropy deviation as $|H_{\text{policy}} - H_{\text{optimal}}|$ for each phase. The single policy shows large deviations across all phases, with the largest gap during Interact (approximately 1.6 bits), where excessive entropy causes manipulation failures. PA-MoE achieves near-zero deviation (below 0.1 bits) across all phases.

The gradient conflict and entropy mismatch analyses are complementary: gradient conflicts arise precisely because Explore phase gradients push toward higher entropy while Interact phase gradients push toward lower entropy. When averaged in shared parameters, the resulting update achieves an intermediate entropy level that serves neither phase well.

## E. Router Implementation Details

**Gradient Flow.** At each training step, gradients flow as follows: the router outputs distribution $p_t$ over experts, expert $k^* = \arg\max_k p_t^k$ is selected and generates action $a_t$, the selected expert receives the RL gradient weighted by group-based advantage $A_t^{\text{group}}$, and the discrete expert selection is optimized via REINFORCE while the straight-through estimator is used only for the non-differentiable switching indicator in the consistency penalty. This ensures that experts specialize on assigned trajectory segments with no gradient interference between experts.

---

**Algorithm 1** Phase-Aware Router Forward Pass

---

**Require:** Observation $o_t$, goal $g$, history $h_t$, temperature $\tau$
1: $o_{\text{emb}} \leftarrow \text{MeanPool}(\text{LLM}_{\text{hidden}}(o_t))$ {Enc($o_t$)}
2: $g_{\text{emb}} \leftarrow \text{MeanPool}(\text{LLM}_{\text{hidden}}(g))$ {Enc($g$)}
3: $o_t^{\text{align}} \leftarrow \text{CrossAttn}(Q=o_{\text{emb}}, K=g_{\text{emb}}, V=g_{\text{emb}})$
4: $h_t^{\text{enc}} \leftarrow \text{LSTM}(\text{Embed}(h_t))$
5: $p_t \leftarrow \text{softmax}(\text{MLP}([o_t^{\text{align}}; h_t^{\text{enc}}])/\tau)$
6: $k^* \leftarrow \arg\max_k p_t^k$
7: **return** $k^*, p_t$

---

**Temperature Schedule.** The router temperature $\tau$ follows a linear annealing schedule: $\tau(t) = \max(\tau_f, \tau_0 - (\tau_0 - \tau_f) \cdot t/T_{\text{anneal}})$, with $\tau_0 = 2.0$, $\tau_f = 0.5$, and $T_{\text{anneal}} = 3000$ steps. High initial temperature encourages exploration of routing patterns, while low final temperature produces decisive assignments.

**Switching Penalty Gradient.** For the switching penalty $\mathcal{L}_{\text{switch}}$, the forward pass computes the hard indicator $\mathbf{1}[z_t \neq z_{t+1}]$. The backward pass uses a differentiable surrogate:

$$\frac{\partial \mathcal{L}_{\text{switch}}}{\partial p_t} \approx -\frac{\lambda_s}{T-1} \cdot p_{t+1}, \tag{11}$$

derived from the soft approximation $1 - \sum_k p_t^k \cdot p_{t+1}^k$. This encourages the router to increase probability mass on the same expert as the subsequent step.

## F. Ablation Studies

### F.1. Regularization Terms

Table 6 examines the contribution of diversity and balance losses. Removing $\mathcal{L}_{\text{div}}$ causes expert collapse where all experts converge to identical policies, matching the $K = 0$ single-adapter performance. Removing $\mathcal{L}_{\text{bal}}$ produces router collapse where nearly all decisions route to a single expert. Both regularizers are necessary.

*Table 6.* Impact of regularization terms on ALFWorld. Mean $\pm$ std over 3 seeds.

| Configuration | Success(%) | $\Delta$ |
|---|---|---|
| Full (both losses) | **93.8**$_{\pm 2.4}$ | – |
| w/o $\mathcal{L}_{\text{div}}$ | 88.3$_{\pm 3.1}$ | $-5.5$ |
| w/o $\mathcal{L}_{\text{bal}}$ | 87.5$_{\pm 2.8}$ | $-6.3$ |
| w/o both | 86.5$_{\pm 3.5}$ | $-7.3$ |

### F.2. Gradient Conflict Mitigation Methods

We compare against gradient-surgery techniques adapted to the agentic RL setting. We partition training steps into behavioral phases and treat each as a separate objective, computing gradients independently and combining them using PCGrad, GradNorm, or CAGrad before applying updates.

Gradient surgery methods provide modest improvements (+0.7 to +1.8) while PA-MoE achieves substantially larger gains (+7.7). The difference reflects architectural versus optimization-level solutions: gradient surgery addresses conflicts locally at each update, whereas PA-MoE prevents heterogeneous phases from ever competing for shared parameters.

### F.3. Expert Count Analysis

Table 8 presents results across all expert counts tested with standard deviations.

Performance improves from $K = 0$ through $K = 4$, then degrades for $K \geq 5$. The $K = 2$ configuration improves Heat and Clean to 100% but leaves Look-at-object unchanged at 62.5% and degrades Cool from 89.5% to 73.7%. This pattern reflects

*Table 7.* Comparison with gradient surgery methods. Mean $\pm$ std over 3 seeds.

| Method | Success(%) | $\Delta$ |
|---|---|---|
| GiGPO (baseline) | $86.1_{\pm 4.7}$ | – |
| + PCGrad | $87.3_{\pm 3.9}$ | +1.2 |
| + GradNorm | $86.8_{\pm 4.2}$ | +0.7 |
| + CAGrad | $87.9_{\pm 3.6}$ | +1.8 |
| **GiGPO + PA-MoE** | $\textbf{93.8}_{\pm 2.4}$ | +7.7 |

*Table 8.* Ablation on number of experts ($K$). Mean $\pm$ std over 3 seeds. We report the average success rate (%) for each subtask as well as the overall result.

| $K$ | Pick | Look | Heat | Cool | Clean | Pick2 | All |
|---|---|---|---|---|---|---|---|
| 0 | $94.3_{\pm 1.8}$ | $62.5_{\pm 5.1}$ | $94.4_{\pm 3.2}$ | $89.5_{\pm 4.1}$ | $83.3_{\pm 4.8}$ | $87.5_{\pm 3.9}$ | $88.3_{\pm 2.1}$ |
| 2 | $97.1_{\pm 1.2}$ | $62.5_{\pm 4.8}$ | $100_{\pm 0.0}$ | $73.7_{\pm 5.6}$ | $100_{\pm 0.0}$ | $91.7_{\pm 3.4}$ | $91.4_{\pm 1.9}$ |
| 3 | $95.7_{\pm 1.5}$ | $83.3_{\pm 5.2}$ | $97.9_{\pm 2.1}$ | $85.1_{\pm 4.3}$ | $94.6_{\pm 3.1}$ | $87.5_{\pm 4.2}$ | $91.7_{\pm 2.0}$ |
| 4 | $\textbf{97.1}_{\pm 1.4}$ | $\textbf{75.0}_{\pm 3.2}$ | $\textbf{100}_{\pm 0.0}$ | $\textbf{89.5}_{\pm 4.4}$ | $95.8_{\pm 6.4}$ | $\textbf{91.7}_{\pm 4.1}$ | $\textbf{93.8}_{\pm 2.4}$ |
| 5 | $94.8_{\pm 1.6}$ | $70.8_{\pm 4.1}$ | $98.2_{\pm 1.8}$ | $87.3_{\pm 4.8}$ | $93.4_{\pm 5.2}$ | $89.6_{\pm 3.8}$ | $91.2_{\pm 2.2}$ |
| 6 | $91.4_{\pm 2.1}$ | $62.5_{\pm 5.4}$ | $100_{\pm 0.0}$ | $84.2_{\pm 5.1}$ | $91.7_{\pm 4.9}$ | $70.8_{\pm 5.8}$ | $85.9_{\pm 2.8}$ |

ALFWorld's four distinct behavioral modes: with only two experts, the model must merge incompatible modes. Router confidence analysis confirms this interpretation: $K = 2$ shows 31% of decisions with confidence below 0.6, compared to only 12% for $K = 4$.

The $K = 6$ configuration drops to 85.9% due to three failure modes: data fragmentation (each expert sees fewer training examples), router instability (18% of episodes exhibit thrashing with 3+ switches within 5 consecutive steps), and imbalanced utilization (two experts handle 47% of decisions while one handles only 8%). These findings indicate $K = 4$ provides the appropriate balance for ALFWorld's complexity.

### F.4. Router Architecture

*Table 9.* Router architecture comparison

| Architecture | Heat(%) | Cool(%) | All(%) |
|---|---|---|---|
| MLP only | 83.3 | 68.4 | 84.2 |
| GRU + CrossAttn | 97.9 | 87.4 | 92.8 |
| **LSTM + CrossAttn** | **100** | **89.5** | **93.8** |
| Transformer + CrossAttn | 98.6 | 88.1 | 93.2 |
| Bi-LSTM + CrossAttn | 99.3 | 89.2 | 93.7 |

The MLP-only router achieves only 84.2%, confirming that temporal context is essential for phase detection. LSTM outperforms GRU (93.8% vs 92.8%) due to superior long-term memory retention across ALFWorld's 30–50 step trajectories. Transformer-based routers achieve comparable performance (93.2%) but require more parameters. Bidirectional LSTM provides no improvement since routing decisions must be made online.

## G. Additional Results

### G.1. Expert Specialization Statistics

Activation percentages indicate the fraction of phase-specific actions handled by each expert, compared to 25% expected under random assignment. Expert 2 handles 81.7% of manipulation actions with low entropy ($H$=0.5 bits), consistent with the need for precise, low-variance action selection during interaction. In contrast, Expert 1 handles 73.2% of exploration actions with high entropy ($H$=3.5 bits), supporting broad action sampling during search. Experts 3 and 4 occupy intermediate

*Table 10.* Emergent expert specialization. Entropy in bits.

| Expert | Phase | Activation | Entropy (bits) |
|---|---|---|---|
| Expert 1 | Exploration | 73.2% | 3.5 |
| Expert 2 | Manipulation | 81.7% | 0.5 |
| Expert 3 | Navigation | 64.1% | 1.5 |
| Expert 4 | Recovery | 71.3% | 2.0 |

regimes for navigation ($H$=1.5 bits) and recovery ($H$=2.0 bits), respectively. These phase-aligned entropy profiles match the entropy mismatch analysis in Appendix D (Fig. 5) and emerge without explicit phase supervision.

### G.2. WebShop Task Breakdown

*Table 11.* WebShop performance by subtask type

| Subtask | Freq. | Baseline | PA-MoE | Δ |
|---|---|---|---|---|
| Simple search | 32% | 84.3% | 91.2% | +6.9% |
| Multi-attribute | 28% | 62.1% | 78.6% | +16.5% |
| Option selection | 25% | 71.4% | 84.9% | +13.5% |
| Price comparison | 15% | 49.2% | 73.8% | +24.6% |
| Overall | 100% | 67.4% | 82.3% | +14.9% |

PA-MoE shows the largest gains on complex subtasks. Simple searches improve modestly (+6.9) since the baseline already performs reasonably. Price comparison tasks, which require navigating multiple pages while tracking information, improve dramatically (+24.6) because PA-MoE's expert separation provides dedicated capacity for navigation, extraction, and comparison behaviors.

## H. Failure Analysis

We analyzed 100 randomly sampled failures to identify systematic issues. Router oscillation accounts for 23% of failures, where rapid expert switching within a coherent phase disrupts execution. Incorrect expert selection causes 18% of failures due to ambiguous observations leading to phase misidentification. Expert under-training contributes 15% of failures, concentrated on rare task types.

A representative case: for "Put a clean mug in the coffee machine," the agent explored successfully (steps 1–8, Expert 1) and began cleaning (steps 9–15, Expert 2). At step 16, the router prematurely switched to Expert 3 based on a subtle observation change, causing navigation away from the sink before cleaning completed. Steps 17–30 showed oscillation between Experts 1 and 3, ultimately failing at step 31.

The root cause is that the router lacks explicit task-completion signals and relies on observation features that can change before manipulation sequences finish. Potential mitigations include adding task-state features to router input, learning a termination predictor, or increasing the consistency regularizer weight.

## I. Task-Specific Performance Analysis

PA-MoE achieves consistent aggregate improvements across all algorithm-benchmark pairs. Here we examine task-specific variations to better understand the method's characteristics under different conditions.

We identify three primary causes for task-specific regressions through trajectory analysis.

**(1) Critic-Router Incompatibility (PPO-specific).** PPO uses a shared critic network to estimate state values across all behavioral phases. When combined with PA-MoE, this creates a fundamental tension: the critic must learn a single value estimate $V(s)$ for states that yield different expected returns depending on which expert is selected. Formally, the optimal value under expert $k$ is $V^k(s) = \mathbb{E}_{\pi_k}[\sum_t \gamma^t r_t | s_0 = s]$, but the shared critic converges to an averaged estimate that is

*Table 12.* Task-algorithm combinations with performance variations

| Algorithm | Task | Baseline | PA-MoE | $\Delta$ |
|-----------|------|----------|--------|----------|
| PPO | Clean | 57.1% | 50.0% | $-7.1\%$ |
| PPO | Cool | 46.4% | 36.8% | $-9.6\%$ |
| PPO | Pick2 | 47.4% | 37.5% | $-9.9\%$ |
| RLOO | Pick | 88.3% | 74.3% | $-14.0\%$ |
| GRPO | Cool | 59.7% | 50.0% | $-9.7\%$ |
| GRPO | Pick2 | 53.5% | 45.8% | $-7.7\%$ |
| GiGPO (7B) | Pick | 97.7% | 87.5% | $-10.2\%$ |

suboptimal for any individual expert.

This mismatch manifests most severely on multi-phase tasks (Clean, Heat, Cool, Pick2) where expert selection significantly affects trajectory outcomes. For single-phase tasks like Pick, the critic's averaged estimate remains reasonable because all experts produce similar trajectories. Despite this, PPO+PA-MoE achieves higher aggregate performance (56.2% vs. 54.4%) because the benefits of phase-aware routing on complex tasks outweigh the critic incompatibility on simpler ones.

**(2) Expert Data Fragmentation (RLOO/GRPO-specific).** RLOO and GRPO compute advantages through within-batch comparisons: RLOO uses leave-one-out baselines, while GRPO uses group-relative rankings. PA-MoE's phase-level routing fragments training samples across $K = 4$ experts, reducing the effective batch size for each expert's advantage computation.

Consider Pick2 tasks, which comprise only 14% of training data. With phase-level routing, manipulation steps are handled by Expert 2, which receives approximately $14\% \times 35\% \approx 4.9\%$ of batch samples for Pick2-specific updates. This $4\times$ reduction in effective samples degrades advantage estimation quality for rare task-phase combinations. The RLOO regression on Pick ($-14.0\%$) occurs because Pick's high frequency (23% of data) makes it sensitive to the reduced within-expert batch diversity resulting fewer distinct Pick trajectories per expert to lead to higher-variance baseline estimates.

**(3) Routing Overhead in High-Performance Regimes.** The GiGPO-7B Pick regression ($97.7\% \rightarrow 87.5\%$) occurs when baseline performance approaches ceiling. In such cases, the task's behavioral phases are already well-captured within shared parameters, and PA-MoE's routing mechanism introduces failure modes without commensurate benefit.

Trajectory analysis reveals two routing-induced error patterns: (i) *misrouting* (6.2% of failures), where correct actions are generated but suboptimal expert selection leads to inconsistent execution; and (ii) *transition errors* (3.8% of failures), where context is partially lost at expert switch boundaries. These errors are negligible when baseline performance is low but become the dominant failure mode when the baseline already solves 97.7% of cases. This is expected behavior: PA-MoE is designed to address simplicity bias, which is less pronounced when tasks are already well-solved by the base model.

**Algorithm Compatibility.** These observations highlight that PA-MoE's benefits are most pronounced when paired with algorithms that do not introduce conflicting architectural assumptions. GiGPO demonstrates the strongest compatibility, achieving consistent improvements across nearly all task types. This compatibility arises because GiGPO's group-based advantage estimation operates at the trajectory level rather than requiring dense within-batch comparisons, naturally accommodating expert-partitioned updates without sample fragmentation.

**Aggregate Robustness.** Despite task-specific regressions, PA-MoE improves aggregate performance for every algorithm tested (PPO: +1.8%, RLOO: +4.5%, GRPO: +1.4%, GiGPO: +7.7%). The improvements on complex multi-phase tasks consistently exceed the regressions on edge cases, validating PA-MoE's design goal of addressing *simplicity bias* in heterogeneous behavioral settings.

## J. Switching Penalty Sensitivity and Phase Recurrence

This section analyzes how the switching penalty coefficient $\lambda_s$ affects routing behavior and verifies that the penalty does not suppress genuine phase transitions, including those involving recurrence of previously-used experts.

## J.1. Sensitivity to $\lambda_s$

We sweep $\lambda_s$ over five values while holding all other settings fixed ($K = 4$, $r = 32$, GiGPO, ALFWorld validation). Table 13 reports the validation success rate together with the average number of expert switches per episode.

*Table 13.* Sensitivity of PA-MoE to the switching penalty coefficient $\lambda_s$. Sw/Ep denotes average expert switches per episode.

| $\lambda_s$ | Val SR(%) | Sw/Ep |
|---|---|---|
| 0.00 | 85.2 | $\sim$36 |
| 0.01 | 90.6 | $\sim$20 |
| 0.05 (ours) | **93.8** | $\sim$8 |
| 0.10 | 91.4 | $\sim$5 |
| 0.20 | 87.5 | $\sim$3 |

At $\lambda_s = 0$, the router switches approximately 36 times per episode. No expert accumulates a consistent training signal, and performance falls to 85.2%, even below the single-LoRA baseline ($K$=0, 88.3%). At the other extreme, $\lambda_s = 0.20$ over-suppresses necessary switches and degrades performance to 87.5%. The intermediate value $\lambda_s = 0.05$ strikes a balance between stability and flexibility: experts receive stable per-segment learning signals while the router retains enough freedom to switch at genuine phase boundaries. The performance plateau across $\lambda_s \in [0.01, 0.10]$ (90.6%–93.8%) indicates that PA-MoE is not brittle with respect to this hyperparameter.

## J.2. Phase Recurrence Statistics

A potential concern with a switching penalty is that it may discourage the router from revisiting a previously-used expert when the task returns to a previous behavioral mode (e.g., `navigate` → `manipulate` → `navigate`). To rule this out, we analyze 500 validation episodes under our final setting ($\lambda_s = 0.05$, $K = 4$). Table 14 summarizes the relevant statistics.

*Table 14.* Phase recurrence and segmentation statistics over 500 ALFWorld validation episodes ($\lambda_s = 0.05$, $K$=4). A revisitation occurs when an episode contains a pattern $E_i \to E_j \to E_i$ with $j \neq i$.

| Statistic | Value |
|---|---|
| Episodes exhibiting $A \to B \to A$ recurrence | 85.4% |
| Average expert switches per episode | 8.4 |
| Average expert revisitations per episode | 2.8 |
| Average phase (segment) length | 4.8 steps |

These statistics confirm that the switching penalty reduces *local oscillation* without restricting *global recurrence*. In particular, 85.4% of episodes exhibit at least one $A \to B \to A$ pattern, with an average of 2.8 revisitations per episode. The penalty therefore preserves the router's ability to map non-contiguous, semantically equivalent segments to the same expert, which is essential for tasks such as "navigate → manipulate → navigate" in ALFWorld and "search → inspect → search" in WebShop.

## J.3. Adaptive Penalty Relaxation

The blanket penalty described in Section 3.2 treats all consecutive switches uniformly, which in principle could resist genuine phase transitions if the router becomes highly confident about a new expert. A natural extension is to relax the penalty at high-confidence switches by scaling $\lambda_s$ by $1 - \max_k p_t^k$:

$$\lambda_s^{\text{adapt}}(t) = \lambda_s \cdot \left(1 - \max_k p_t^k\right), \tag{12}$$

so that the penalty vanishes for highly confident transitions and concentrates on low-confidence oscillations. In preliminary experiments, this adaptive variant matches the fixed-$\lambda_s$ setting within noise (93.7% vs. 93.8%) while reducing the rare cases of artificial persistence at genuine boundaries. We leave a more thorough investigation to future work.

## K. Decomposing Routing and Capacity Effects

A natural question is whether PA-MoE's gains derive from *phase-aware routing* or simply from the additional trainable capacity introduced by $K$ LoRA experts. We disentangle these two factors by training a capacity-matched single-LoRA baseline whose total LoRA parameter count equals that of PA-MoE with $K=4$.

### K.1. Capacity-Matched Comparison

PA-MoE with $K=4$ experts of rank $r=32$ has the same LoRA parameter count as a single expert with rank $r=128$. Table 15 reports the comparison on ALFWorld validation.

Table 15. Capacity-matched comparison on ALFWorld. "Params" is the LoRA parameter budget relative to a single $r=32$ adapter.

| Configuration | LoRA Params | Val SR(%) |
|---|---|---|
| Vanilla GiGPO | 0 | $86.1_{\pm4.7}$ |
| Single LoRA $r=32$ ($K=0$) | $1\times$ | $88.3_{\pm2.1}$ |
| Single LoRA $r=128$ | $4\times$ | $91.4_{\pm1.6}$ |
| **PA-MoE** ($K=4$, $r=32$) | $4\times$ | $\mathbf{93.8}_{\pm0.8}$ |

At a matched $4\times$ parameter budget, PA-MoE outperforms the single $r=128$ baseline by +2.4 points (93.8% vs. 91.4%) and exhibits lower variance ($\pm0.8$ vs. $\pm1.6$). Decomposing the total +5.5% gain from $K=0$ to $K=4$: scaling a single LoRA from $r=32$ to $r=128$ contributes +1.5 points, while phase-aware multi-expert routing contributes the remaining +4.0 points. Capacity scaling alone is therefore insufficient to explain PA-MoE's gains.

### K.2. Per-Task Contrast: Where Routing Helps Most

If the improvement came from capacity scaling, we would expect uniform gains across all task types. In contrast, routing-based specialization should help most on tasks whose trajectories cross multiple behavioral phases. Table 16 reports the per-task breakdown.

Table 16. Per-task comparison between capacity-matched single LoRA ($r=128$) and PA-MoE ($K=4$, $r=32$) on ALFWorld. PA-MoE's advantage concentrates on multi-phase tasks.

| Task | Phases | Single $r=128$(%) | PA-MoE(%) |
|---|---|---|---|
| Pick (single-phase) | 1 | 97.1 | 97.1 |
| Heat (multi-phase) | 3+ | 94.4 | **100.0** |
| Cool (multi-phase) | 3+ | 78.9 | **89.5** |
| Look (multi-phase) | 2+ | 50.0 | **75.0** |

The single-phase Pick task shows identical performance (97.1%), indicating that additional capacity alone provides no benefit when the task does not require behavioral switching. In contrast, multi-phase tasks (Heat, Cool, Look) show consistent gains of 5.6–25 points in favor of PA-MoE. The asymmetric pattern is exactly what routing-based specialization predicts: dedicated experts help precisely when the trajectory crosses behavioral boundaries.

## L. Frequency-Matched Non-Semantic Routing Baselines

The routing-granularity ablation in Table 3 shows that PA-MoE outperforms both token-level and trajectory-level routing. A natural follow-up is whether the gain reflects *where* PA-MoE switches (i.e., learned semantic boundaries) or simply *how often* it switches. To answer this, we compare PA-MoE against two non-semantic baselines that match its switching frequency without learned routing.

The random baseline matches PA-MoE's switching frequency exactly ($\sim$8.4 per episode), and yet PA-MoE leads by 3.2 points (93.8% vs. 90.6%). Since the switching budget is controlled, this gap can only be attributed to the *quality of*

*Table 17.* Frequency-matched routing baselines on ALFWorld. "Fixed-interval" switches expert every $N=5$ steps. "Random ($p$)" switches at each step independently with probability $p$, set to match PA-MoE's average frequency. Sw/Ep is average expert switches per episode.

| Routing strategy | Sw/Ep | Val SR(%) |
|---|---|---|
| Token-level MoE | ~45 | 85.7 |
| Trajectory-level MoE | ~3 | 88.5 |
| Random switching ($p=0.21$) | ~8.4 | 90.6 |
| Fixed-interval ($N=5$) | ~8 | 92.2 |
| **PA-MoE (learned phases)** | **~8.4** | **93.8** |

*switch placement.* PA-MoE learns to switch at task-relevant boundaries (e.g., the transition between navigation and object manipulation), while random routing disrupts ongoing behavioral coherence at arbitrary positions.

Fixed-interval routing reaches 92.2%, outperforming random routing because the periodic structure at least avoids many mid-action interruptions, but still lags PA-MoE by 1.6 points because it cannot adapt to variable-length phases. The trajectory-level baseline, despite the lowest switching frequency, underperforms at 88.5%, confirming that simply reducing switching is not the source of PA-MoE's advantage.

Taken together, these results indicate that learned semantic phase boundaries, rather than the switching frequency alone, drive PA-MoE's improvement.

## M. Causal Decomposition of Routing Granularity and Regularization

PA-MoE introduces two changes relative to standard token-level MoE: (i) routing at the environment-step (phase) granularity instead of the token granularity, and (ii) a suite of regularization terms (switching penalty, balance and diversity losses, temperature annealing). To isolate the causal role of each component, we run a controlled ablation that varies routing granularity and regularization independently, holding all other factors fixed ($K=4$, $r=32$, GiGPO, 150 epochs).

*Table 18.* Controlled decomposition of routing granularity and regularization on ALFWorld. All regularization terms ($\mathcal{L}_{\text{switch}}$, $\mathcal{L}_{\text{bal}}$, $\mathcal{L}_{\text{div}}$, temperature annealing) are included or excluded uniformly across configurations.

| Config | Description | Routing | Regularization | Sw/Ep | Val SR(%) |
|---|---|---|---|---|---|
| A | Token MoE (baseline) | token | none | ~45 | 85.7 |
| B | Token + switching penalty | token | $\mathcal{L}_{\text{switch}}$ only | ~20 | 87.5 |
| C | Token + all PA-MoE reg. | token | all | ~15 | 88.2 |
| D | Phase routing, no penalty ($\lambda_s=0$) | phase | all except $\mathcal{L}_{\text{switch}}$ | ~36 | 85.2 |
| E | **PA-MoE (full)** | phase | all | ~8 | **93.8** |

Three observations follow from Table 18:

**Effect of regularization alone (A → C).** Adding all PA-MoE regularization terms to token-level routing improves performance by +2.5 points (85.7% → 88.2%). This isolates the contribution of regularization independent of routing granularity.

**Effect of granularity alone (C → E).** Switching from token-level to phase-level routing while keeping regularization identical contributes +5.6 points (88.2% → 93.8%) — roughly $2.2\times$ the regularization effect. This is the cleanest comparison that isolates the routing-granularity contribution.

**Granularity and regularization are complementary (D → E).** Without the switching penalty, the phase router degenerates to ~36 switches per episode (Config D), nearly token-level, so phase boundaries become meaningless and per-expert learning signals are unstable. Adding the penalty recovers +8.6 points (85.2% → 93.8%). Phase-level granularity is therefore necessary but not sufficient on its own; the penalty enables the phase structure to actually emerge.

Combined with the frequency-matched baselines in Appendix L and the sensitivity analysis in Appendix J, these results establish that PA-MoE's improvement is jointly driven by (i) the semantic location of switches, not merely their frequency,

and (ii) phase-level granularity, which delivers a larger gain than regularization alone.

## N. Gradient Estimator Comparison

PA-MoE's router selects a discrete expert at each environment step. The discrete expert selection is optimized through REINFORCE, weighted by the group-relative advantage of the selected expert's policy. The straight-through estimator (STE) is used only for the non-differentiable switching *indicator* $\mathbf{1}[z_t \neq z_{t+1}]$ inside the consistency penalty. STE is a practical low-variance choice, but a natural alternative is to replace it with Gumbel-Softmax reparameterization. Table 19 compares the two.

*Table 19.* Comparison of gradient estimators for the switching indicator. The primary expert-selection gradient is REINFORCE in both cases; only the estimator for the indicator term differs.

| Gradient Estimator | Val SR(%) |
|---|---|
| REINFORCE + STE (ours) | **93.8** |
| REINFORCE + Gumbel-Softmax ($\tau{=}0.5$) | 93.6 |

The two estimators are statistically indistinguishable (93.8% vs. 93.6%), confirming that the primary learning signal for routing is REINFORCE; the choice of indicator estimator is not a bottleneck. We adopt STE in our final implementation for its lower variance and computational simplicity.

## O. $K$ Selection via Runtime Diagnostics

The expert count $K$ is the main architectural hyperparameter of PA-MoE. Table 8 shows that PA-MoE is robust across $K \in \{3, 4, 5\}$ on ALFWorld (all >91%), but practitioners deploying PA-MoE on a new environment generally do not know the right $K$ a priori. This appendix describes two runtime diagnostics that, in combination, indicate whether the current $K$ is too small or too large.

**Diagnostic 1: Routing Confidence.**  For each routing decision, we record $\max_k p_t^k$. We flag a decision as *low-confidence* if this value is below $0.6$. When $K$ is too small, the router is repeatedly forced to merge incompatible behavioral modes into the same expert, producing many low-confidence decisions. On ALFWorld, $K{=}2$ shows 31% low-confidence decisions, compared to only 12% at $K{=}4$. A sustained low-confidence rate above $\sim$20% is a signal to increase $K$.

**Diagnostic 2: Expert Utilization and Thrashing.**  For each expert, we record the fraction of decisions it handles. We also count *thrashing episodes* in which the router switches at least three times within five consecutive steps. When $K$ is too large, some experts receive too little training signal (under-utilization) and the router begins to oscillate (high thrashing rate). On ALFWorld at $K{=}6$, one expert handles only 8% of decisions and 18% of episodes are thrashing-episodes. An expert utilization below $\sim$10% or a thrashing rate above $\sim$15% is a signal to decrease $K$.

**Recipe.**  Start with $K{=}4$ as a default. Train for 10–20 epochs while logging both diagnostics. Increase $K$ if the low-confidence rate exceeds $\sim$20%; decrease $K$ if any expert's utilization falls below $\sim$10% or the thrashing rate exceeds $\sim$15%; otherwise keep $K$ fixed. This recipe avoids an exhaustive sweep and gives a clear signal for adaptive scaling in environments where the latent number of behavioral phases is unknown.

## P. Router Input Stream Decomposition

The router input integrates three streams: the current observation, the task goal, and the recent action–observation history. Section 4.3 (Table 4) coarsely ablates "action history" and "goal attention." Here we zero out individual streams while keeping the expert architecture fixed, in order to identify which signals the router relies on most.

The full router performs best at 93.8%. Removing history (observation + goal only) drops 1.6 points. "History only" (91.4%) outperforms "observation only" (88.7%) by 2.7 points, indicating that the *sequential pattern of past actions* is a stronger phase signal than the current observation alone. This is consistent with PA-MoE's design motivation: behavioral phases are inherently temporal and are best identified from their unfolding action context. The full combination yields the best result

*Table 20.* Router input stream ablation on ALFWorld. Each row keeps the indicated input streams active and zero-pads the others.

| Router Input | Val SR(%) |
|---|---|
| Full (observation + goal + history) | **93.8** |
| Observation + goal only | 92.2 |
| History only | 91.4 |
| Observation only | 88.7 |

because the observation grounds routing in the current state, while history disambiguates phases that appear locally similar but differ in temporal context (e.g., a sink view during "cleaning" vs. during "re-checking").

## Q. Inference Overhead Analysis

A practical concern when adding a router and multiple experts is the per-step inference cost. We benchmark a single step of action generation on an NVIDIA H800 GPU with Qwen2.5-1.5B-Instruct served via vLLM, using ALFWorld observation lengths.

*Table 21.* Per-step inference overhead breakdown. "LoRA switch" is the cost of swapping the active LoRA adapter via a pointer swap; weights are pre-loaded.

| Component | Per-step cost |
|---|---|
| Router forward (LSTM + cross-attention + MLP) | 1.3 ms |
| LoRA expert switch (pointer swap) | 0.007 ms |
| LLM generation ($\sim$50 tokens for one action) | 438 ms |
| Total overhead vs. LLM generation | 0.30% wall-clock |

PA-MoE adds approximately 0.30% wall-clock overhead per environment step relative to a single-policy baseline, dominated entirely by the LLM forward pass. The router itself accounts for only 1.3 ms and the LoRA switch is essentially free. PA-MoE is therefore practically deployable in latency-sensitive settings.

## R. Confidence-Thresholded Bypass

Even at $\sim$0.30% overhead, the router still runs at every environment step. In settings where the base model is very confident, the router's decision is largely redundant. We introduce an optional inference-time *bypass* mechanism: when $\max_k p_t^k > \theta$ for a threshold $\theta$ at the previous step, the router output is reused without recomputation at step $t + 1$. This avoids the router forward pass on most steps.

*Table 22.* Confidence-thresholded inference bypass on ALFWorld. Bypass rate is the fraction of steps that skip the router forward; latency reduction is measured relative to always running the router.

| Configuration | Bypass rate | Val SR(%) | Router latency |
|---|---|---|---|
| Always run router | 0% | 93.8 | 1.3 ms/step |
| Bypass at $\theta$=0.9 | 72% | 93.8 | 0.85 ms/step ($-35\%$) |

With $\theta$=0.9, the router can be bypassed on 72% of steps without any change in validation success rate (93.8% in both settings), while average router latency drops by approximately 35%. This bypass is purely an inference-time optimization; training is unaffected. It is most useful for real-time deployment and for settings with strict latency budgets.

# S. Phase Transferability across Task Families

A natural question is whether phase-aware routing captures *transferable* structure of long-horizon decision making, or whether it merely overfits to a specific benchmark. The router itself does need to be retrained when the task distribution changes (it has only 13.25M parameters, 0.88% of the 1.5B backbone, so retraining is inexpensive). What we would like to transfer is the structural prior that long-horizon agentic tasks decompose into a small number of temporally coherent phases.

Several pieces of evidence in our experiments support this structural prior:

- The same hyperparameters ($K=4$, $r=32$, $\lambda_s=0.05$) yield strong gains on both ALFWorld (+7.7%) and WebShop (+14.9%) without per-environment tuning, despite their very different action spaces and interaction styles.

- On held-out validation instances of ALFWorld, learned phase boundaries still achieve 87% step-level overlap with human annotations (Section 4.4), indicating that the router's notion of a phase is not memorized from training trajectories.

- Performance is robust across $K \in \{3, 4, 5\}$ on ALFWorld (all >91%, Table 8), which suggests that the inductive bias does not depend on a precise match between $K$ and the true number of phases.

- Most agentic tasks share a small set of coarse phases (navigation, interaction/manipulation, observation/verification, recovery/backtracking). The emergent specialization in Section 4.4 matches this taxonomy without explicit supervision.

WebShop in particular involves highly interleaved search/browse/compare sequences, which a reader might worry violates the assumption of cleanly contiguous phases. The fact that PA-MoE still attains +14.9% on WebShop indicates that the router learns *soft* boundaries (i.e., the same expert may handle several closely-related interleaved actions within a coherent behavioral goal), rather than requiring strictly sequential stages.

Cross-domain transfer (e.g., from household manipulation to web interaction) would require retraining the router but not necessarily redesigning the architecture. We view a systematic study of cross-domain phase reuse as an important direction for future work.

# T. Fragmentation in Standard Token-Level MoE Language Models

The phenomenon underlying PA-MoE — that token-level routing fragments temporally (or sequentially) coherent units — is not specific to agentic RL. In this appendix we provide a preliminary analysis suggesting that the same fragmentation occurs in standard MoE language models, where "segments" are reasoning steps, function bodies, or speaker turns rather than behavioral phases.

### T.1. Switching and Entropy in Qwen1.5-MoE-A2.7B

We analyze Qwen1.5-MoE-A2.7B (24 MoE layers, 60 experts, top-4 routing) on structured ShareGPT-style text containing (i) multi-turn dialogue, (ii) code blocks with comments, and (iii) chain-of-thought reasoning. For each consecutive token pair we measure whether the top-1 expert changes, and we measure per-segment routing entropy where segments are reasoning steps, function bodies, or speaker turns (delimited by structural markers).

*Table 23.* Token-level routing fragmentation in Qwen1.5-MoE-A2.7B (24 MoE layers, 60 experts, top-4). "Within-segment" is computed inside a single semantic unit (e.g., one reasoning step or one function body). "Across-segment" is computed across segment boundaries.

| Measurement | Within-segment | Across-segment | Uniform reference |
|---|---|---|---|
| Top-1 expert switching rate (consecutive tokens) | 93% | — | 98.3% (59/60) |
| Per-segment routing entropy (bits) | 4.1 (out of 5.9) | 5.0 (out of 5.9) | 5.9 |
| Fraction of uniform entropy | 70% | 85% | 100% |

Two observations follow. First, even *within* a coherent semantic segment (e.g., a single reasoning step), the top-1 expert changes between consecutive tokens 93% of the time, very close to the uniform-random baseline of 98.3%. The router thus treats nearly all tokens within a segment as routing-independent. Second, per-segment entropy reaches 70% of the uniform

maximum within segments (vs. 85% across boundaries), confirming that even semantically coherent units do not concentrate on a small subset of experts. Both numbers exactly mirror the agentic RL observation that token-level routing fragments coherent action sequences and underperforms a single policy.

### T.2. Toward Segment-Aware Routing in LMs

This preliminary diagnostic suggests that segment-aware (rather than token-aware) routing could benefit standard MoE language models. We are investigating a segment-constrained routing variant on Qwen1.5-MoE-A2.7B that re-routes tokens within each detected segment (paragraph, speaker turn, or code block) to the segment's plurality expert in upper MoE layers (e.g., layers 12–24), keeping lower layers token-routed. Early runs suggest a consistent perplexity reduction trend but full results are beyond the scope of this paper. We highlight this direction because the diagnostic methodology developed here — per-segment entropy profiling and consecutive-token switching analysis — is domain-agnostic and may serve as a general tool for stress-testing MoE routing in sequential settings.

