# OpenReview forum: "Phase-Aware Mixture of Experts for Agentic Reinforcement Learning"
_ICML.cc/2026/Conference — ICML 2026 regular_

### Official Review · Reviewer_5XwA · 2026-02-23

**Soundness:** 2
**Presentation:** 3
**Significance:** 3
**Originality:** 3
**Overall Recommendation:** 5
**Confidence:** 3

**Summary:**

The paper points at a compelling limitation of token-level Mixture of Experts (MoE) in agentic RL: excessive expert switching that fragments temporally coherent behavioral patterns. The authors argue that this hinders sequential decision-making and propose Phase-Aware MoE (PA-MoE), which forces temporal consistency through a switching penalty and history-aware router. Experts are implemented as LoRA adapters on a frozen backbone.

Empirically, PA-MoE demonstrate consisten improvements over non-MoE methods accross multiple RL algorithms on ALFWorld and Webshop, with strong ablations on expert count, routing granularity, router architecture and regularization terms.

**Compliance With Llm Reviewing Policy:**

Affirmed.

**Final Justification:**

Rebuttal addressed my important concerns as explained below.

**Key Questions For Authors:**

**1. Causal Role of Expert Switching.** The paper frames excessive expert switching as the central mechanism underlying the degradation of token-level MoE, yet multiple architectural and regularization components are introduced simultaneously. Could the authors more cleanly isolate the causal role of switching by controlling for other factors? For example, what is the performance of phase-level routing without the switching penalty, or token-level routing augmented with the same switching regularization and temporal modeling? Additionally, is there a measurable relationship between switching frequency and performance across hyperparameter sweeps or runs?

**2. Baseline Fairness and Architectural Parity.** It would be helpful to clarify whether the token-level MoE baseline receives comparable architectural capacity and regularization. Specifically, does it benefit from the same balance and diversity losses, temperature annealing, and temporal modeling components as PA-MoE? If token-level routing were combined with the same history-aware router and regularization strategy, would the performance gap persist?

**3. Role of LoRA-Based Experts vs Full Subnetworks.** Experts are implemented as LoRA adapters on a shared frozen backbone, which differs from canonical MoE formulations using independent subnetworks. To what extent are the observed gains tied to low-rank modularization rather than routing granularity? Have the authors evaluated sensitivity to LoRA rank, or compared against fully parameterized expert subnetworks?

**4. Broader Applicability Beyond Agentic RL.** The proposed phase-aware routing is motivated by sequential decision-making in agentic RL, but its broader architectural implications remain speculative. Can the authors provide evidence that similar switching-induced fragmentation exists in standard MoE settings (e.g., language modeling), and that phase-level routing improves specialization or stability there as well?

**Limitations:**

yes

**Strengths And Weaknesses:**

The paper tackles an important issue and presents strong empirical results. However, the experimental evidence does not fully isolate the central claimed mechanism (that expert switching is the primary cause of performance degradation) from other architectural and regularization changes in PA-MoE.

## Soundness
### Strengths
- The mismatch between token-level MoE and temporally extended decision-making is clearly identified.
- A concept of "behavioural phase" is cleanly formalized, with an intuitive switching metric and entropy analysis.
- Very strong empirical evaluation across two benchmarks (ALFWorld, WebShop) and two model sizes (1.5B, 7B) and multiple RL algorithms.
- Extensive ablations (num. experts, routing granularity, router compnents, regularizations terms...)
- A human-labelled phase overlap strengthens the behavioural interpretation.
### Weaknesses
- The central mechanism is not causally isolated. The paper frames excessive expert switching as the core problem, but it remains unclear whether gains stem from either: reduced switching, better router modelling, additional regularization, LoRA/based modularization or general stabilisation, or perhaps a combination of all.
- It is unclear whether the token-level MoE baseline benefits from comparable tuning and regularization. For example, does token-level MoE receive balance/diversity regularization? Would token-level routing combined with temporal modeling narrow the gap? Is switching penalty the dominant factor, or is routing granularity secondary?
- Experts are implemented as LoRA adapters rather than independent subnetworks. While this is efficient, it diverges from canonical MoE formulations. The paper does not discuss whether full subnets would behave differently and how much of the gains are specific to low-rank adaptation.
- There is an important negative result, that token-level MoE does not outperform single-policy baselines. This is an important and potentially impactful negative result, but it is not explored across model scales and task types.
- The methodology for identifying and validating behavioral phases is strong and could have broader diagnostic value for MoE systems beyond this setting. However, this is not properly emphasized and classified as a standalone contribution.

## Presentation
### Strengths
- Clearly written and logically structured, with high detail on experimental validation and extensions in the appendix.
### Weaknesses
- Fig. 2 switching metric requires clearer contextualization (how it is computed, for which environments, if there is convergence parity, architectural and design fairness...).
- Fig. 3 looks more conceptual and cartoon-like rather than engineering-rigorous.
- The behavioural phase diagnostic tool is under-emphasised.

## Significance
### Strengths
- Addresses a real and relevant mismatch between token-level MoE and sequential decision-making.
- Demonstrates parameter efficiency.
- Shows consistent and substantial improvements across multiple RL algorithms.
- Introduces a behavioural phase consistency concept that is relevant to stress-test MoE in sequential decision-making problems.
### Weaknesses
- If the improvements stem from better regularization rather than routing granularity, significance narrows.
- The broader applicability to standard MoE remains speculative.

## Originality
### Strengths
- The phase-level routing for RL agents appears novel and is clearly contrasted with token-level MoE, trajectory-level MoE and gradient surgery methods.
- The formalization of the entropy mismatch across behavioral phases is interesting, and the human-aligned phase overlap is a novel diagnostic element.
### Weaknesses
- Re-combines known components (LoRA, LSTM router, switching penalty and balance/diversity losses).
- As currently framed, innovation lies in the integration rather than new learning principles.

**Score:** I consider this a Weak Accept (4) at its current stage; however, if the authors can more clearly isolate and substantiate the causal role of expert switching relative to the other architectural and regularization components, this work would merit an Accept (5). A strong accept (6) could be considered if the broader applicability of phase-aware routing to standard MoE settings (beyond agentic RL) were demonstrated.

---
> **After rebuttal**
> If the newly reported results are added to the revised paper, my score is raised to 5 (Accept).

---

> ### Author Rebuttal · Authors · 2026-03-30
>
> We are grateful for the reviewer's insightful comments.
>
> **Q1. Causal decomposition of expert switching.**
>
> We run a controlled ablation that varies routing granularity and regularization independently, holding everything else fixed (K=4, r=32, GiGPO, 150 epochs).
>
> |Config|Routing|Sw/Ep|Val SR(%)|
> |---|---|---|---|
> |A. Token MoE (paper)|token|~45|85.7|
> |B. Token + penalty|token|~20|87.5|
> |C. Token + all PA-MoE reg|token|~15|88.2|
> |D. Phase, $\lambda_s$=0|phase|~36|85.2|
> |**E. PA-MoE (full)**|**phase**|**~8**|**93.8**|
>
> Adding all regularization to token-level routing (A→C) yields +2.5 points. Switching from token to phase granularity while keeping regularization identical (C→E) yields +5.6 points—2.2× the regularization effect. Within phase routing, the penalty contributes +8.6 points (D→E), showing that both components are needed and complement each other.
>
> Note that Config D (phase, no penalty) slightly underperforms Config A (token, no penalty). Without the penalty, the phase router switches 36 times per episode, nearly as often as token-level MoE (~45), so phase boundaries become meaningless and the router cannot accumulate stable per-expert learning signals. The penalty is what enables the phase structure to emerge. This is why C→E (both with identical regularization, differing only in granularity) is the correct comparison, yielding a +5.6 point gap that isolates the granularity contribution.
>
> Additional results across different values of $\lambda_s$ (response to Reviewer 4BvN, W1&Q1) show that performance improves at first but deteriorates as $\lambda_s$ increases further, directly implicating switching frequency as a causal variable. Frequency-matched non-semantic baselines (fixed-interval 92.2%, random 90.6%) lag PA-MoE by 1.6–3.2 points, isolating the contribution of learned switch placement.
>
> **Q2. Baseline fairness.**
>
> Config C gives token-level routing all PA-MoE components (same router, penalty, balance/diversity losses, annealing), differing only in granularity. Even so, token-level reaches only 88.2%, 5.6 points below PA-MoE, because per-token switching fragments within-action coherence regardless of regularization quality. Standard token-level MoE (85.7%) even underperforms the single-policy baseline (88.3%), confirming token-level switching actively harms agentic performance.
>
> **Q3. LoRA rank sensitivity.**
>
> At matched 4× parameter budget (response to Reviewer 4BvN, W2&Q3), PA-MoE K=4 r=32 (93.8±0.8) outperforms single LoRA r=128 (91.4±1.6) by +2.4 points with lower variance.
>
> **Q4. Broader applicability.**
>
> *Fragmentation in standard MoE.* We analyze Qwen1.5-MoE-A2.7B (24 MoE layers, 60 experts, top-4) on structured ShareGPT text covering multi-turn dialogue, code-with-comments, and chain-of-thought reasoning. Within semantically coherent segments such as a single reasoning step or function body, the top-1 expert switching rate is 93% per consecutive token pair, close to the uniform-random baseline of 98.3% (59/60). The per-segment expert entropy reaches H=4.1 out of a maximum 5.9 bits (70% of uniform), compared to H=5.0 bits across segment boundaries (85% of uniform). This confirms that standard MoE routers treat within-segment tokens nearly independently, directly mirroring the agentic fragmentation where token-level MoE (85.7%) underperforms even a single policy (88.3%).
>
> *Generality of the mechanism.* Our causal ablation (Q1) establishes that aligning routing boundaries with semantic structure, rather than merely reducing switching frequency, drives a +5.6 point gain, and frequency-matched non-semantic baselines still lag by 1.6–3.2 points. This principle is domain-agnostic, as the granularity mismatch between token-level routing and segment-level semantics is identical whether segments are behavioral phases in RL or reasoning steps in language modeling. The diagnostic methodology we develop (per-segment entropy profiling, switching frequency analysis) applies directly to any MoE architecture.
>
> *Segment-aware routing in LM.* We are implementing a segment-constrained routing variant on Qwen1.5-MoE-A2.7B that re-routes tokens within each detected segment (e.g., paragraph, speaker turn, or code block) to the segment’s plurality expert in layers 12–24. Preliminary runs suggest a consistent perplexity reduction trend. Due to the rebuttal timeline, full results are not yet available, but we will provide complete results during the discussion period.
>
> **Presentation.** We improve Fig. 2 with computation details and convergence parity, Fig. 3 with gradient flow, and elevate the phase diagnostic as a methodological contribution.
>
> We would be grateful if the reviewer could reconsider the score in light of these results.

---

> > ### Author Rebuttal · Reviewer_5XwA · 2026-03-31
> >
> > My concerns are addressed by the authors' rebuttal. If the newly reported results are added to the revised paper, my score is raised to 5 (Accept).

---

> > > ### Author Response · Authors · 2026-04-04
> > >
> > > We sincerely thank the reviewer for raising the score and for the constructive engagement throughout this discussion. We will incorporate all promised revisions in the camera-ready version.

---

### Official Review · Reviewer_z4xY · 2026-03-11

**Soundness:** 2
**Presentation:** 2
**Significance:** 2
**Originality:** 3
**Overall Recommendation:** 4
**Confidence:** 4

**Summary:**

This paper identifies what I consider to be a highly relevant issue in agentic reinforcement learning: the inherent limitation of relying on a single policy network to handle both simple and complex tasks effectively. The authors argue that such a unified policy tends to exhibit *simplicity bias*, where frequent and easy tasks dominate gradient updates, leaving insufficient representational capacity for more challenging, multi-step behaviors.  To address this problem, the paper proposes a Phase-Aware Mixture-of-Experts (PA-MoE) framework. The key idea is to partition an agent’s trajectory into behaviorally coherent phases and assign different expert policies to different phases. Instead of adopting conventional token-level routing, the proposed approach introduces a phase-level router that learns latent phase boundaries directly from the RL objective. This design better aligns expert specialization with the temporal structure of sequential decision-making. To further enhance performance, the authors introduce a Temporal Consistency mechanism that discourages frequent expert switching. By incorporating a switching penalty and temperature annealing, the router is encouraged to produce temporally stable and phase-consistent assignments, thereby maintaining coherence within each behavioral segment while still allowing adaptation when necessary. Experimentally, the paper evaluates PA-MoE on two representative benchmarks, ALFWorld and WebShop. The authors combine PA-MoE with several policy optimization algorithms, including PPO, RLOO, and GRPO. The empirical results demonstrate that PA-MoE integrates smoothly with these algorithms and achieves improved performance on most tasks, particularly those requiring complex multi-step interactions. Overall, the experimental findings provide convincing evidence that PA-MoE is a flexible and effective architectural enhancement for agentic reinforcement learning.

**Compliance With Llm Reviewing Policy:**

Affirmed.

**Final Justification:**

My concerns have been addressed, and I have raised my score.

**Key Questions For Authors:**

1. Is the phase segmentation learned by the router domain-specific? In other words, if the task distribution changes (e.g., moving to a different environment or a different type of agentic task), will the learned phase boundaries still generalize, or does the router need to be retrained from scratch? It would be helpful if the authors could clarify whether phase-aware routing captures transferable structural patterns or mainly overfits to the current benchmark.

2. During inference, how does the PA-MoE execution mechanism affect latency? Since each step requires routing and selecting among multiple experts (even if only one expert is activated), there is presumably some additional computational overhead compared to a single-policy model. A quantitative analysis of inference-time cost, including routing overhead and potential impact on response time, would strengthen the practical evaluation of the method.

**Limitations:**

Please refer to the weaknesses discussed above, which outline the main limitations of the proposed method, including the ambiguity of phase definition, indirect and potentially unstable routing supervision, possible sample inefficiency due to isolated expert updates, limited routing capacity, lack of knowledge sharing across experts, additional computational overhead, and the absence of rigorous theoretical analysis.

**Strengths And Weaknesses:**

## Strengths

1. I think the motivation of this paper is very clear. It points out a genuinely important issue in agentic reinforcement learning: most existing work focuses on improving the performance of a single policy, but rarely questions whether relying on a single policy network is itself fundamentally limiting. The discussion on simplicity bias is intuitive and well-argued, and I find the problem setting both reasonable and meaningful.

2. The proposed PA-MoE framework is relatively simple in design but practical in implementation. It can be combined with different RL algorithms without major modification and consistently brings performance gains. The fact that it can be incorporated in a plug-and-play manner is a strong advantage, and its flexibility makes it appealing for broader adoption.

3. The experimental evaluation is fairly comprehensive. The authors choose two representative benchmarks, ALFWorld and WebShop, covering both household manipulation tasks and shopping/navigation scenarios. They also compare against multiple baselines (e.g., PPO, RLOO, GRPO), which makes the empirical claims more convincing.

4. The paper includes sufficient ablation studies. Different components of PA-MoE, such as routing design and regularization terms, are carefully analyzed. These ablations help justify the design choices and make the overall contribution clearer.


## Weaknesses

1. Although the paper formally defines “phase” in Definition 3.1, I find the notion still somewhat ambiguous. In multi-step tasks, different phases are usually associated with semantically meaningful stages such as planning, searching, or interacting. However, in this work, a phase is defined simply as a contiguous segment assigned to the same expert. It remains unclear whether such routing-induced phases truly correspond to meaningful task stages [1]. Since phase boundaries are entirely determined by the router, without explicit semantic or structural constraints, incorrect routing decisions would directly lead to incorrect phase segmentation. The paper does not fully address whether the learned phases are consistently aligned with interpretable task substructures.

2. The router does not directly optimize phase segmentation; instead, it receives gradients indirectly through the RL loss via expert selection. Given that RL objectives are often noisy and high-variance, especially in long-horizon tasks, the supervision signal for learning phase boundaries may be weak. I am concerned that the router might struggle to reliably learn meaningful phase distinctions under such indirect and potentially unstable signals.

3. At each step, only the expert selected by the router is updated. This design raises two potential issues. First, early routing mistakes may cause updates to be applied to suboptimal experts, while other experts receive no gradient signal, potentially introducing bias that persists throughout training. Second, updating only one expert per step may reduce sample efficiency and prolong training time. The paper does not provide a detailed discussion of these trade-offs.

4. The router is implemented as a relatively small 3-layer LSTM with an MLP head. Compared to the base LLM, this module is lightweight. It is not entirely convincing that such a small network can reliably identify complex behavioral phases, especially in long episodes where dependencies may span many timesteps. The scalability of this routing mechanism to more complex environments remains questionable.

5. Experts are implemented as independent LoRA adapters. While this enforces parameter isolation, it also prevents knowledge sharing across experts. As a result, useful behaviors learned by one expert cannot be directly reused by others, which may reduce learning efficiency and introduce redundancy across experts [2,3]. Prior sparse expert systems emphasize routing stability, load balancing, and controlled expert utilization to mitigate collapse or redundancy. In contrast, strict LoRA isolation may limit cross-phase transfer and efficient parameter reuse.

6. Although LoRA reduces per-expert parameter cost, the framework still requires maintaining multiple experts and a router. This inevitably introduces additional overhead during both training and inference. The paper does not provide a detailed analysis of computational cost, memory footprint, or latency. In addition, the switch penalty is treated as a manually tuned hyperparameter, but its sensitivity and impact on performance are not thoroughly analyzed.

7. The empirical results are strong, but the method lacks deeper theoretical grounding. While some intuitive arguments are provided (e.g., gradient isolation), the paper does not offer a rigorous analysis of why phase-aware routing should consistently mitigate simplicity bias under general conditions [2]. Prior work on sparse expert architectures provides more detailed discussion of routing stability and transferability, which is currently missing here.

8. Figure 3, which illustrates the PA-MoE architecture, is relatively coarse. It does not clearly convey the full execution flow, particularly how the router selects experts across phases and how gradients propagate through the system. A more detailed diagram would improve clarity and help readers better understand the overall mechanism.

---

### References

- [1] Goyal, Anirudh, et al. "Recurrent independent mechanisms." arXiv preprint arXiv:1909.10893 (2019).

- [2] Zoph, Barret, et al. "St-moe: Designing stable and transferable sparse expert models." arXiv preprint arXiv:2202.08906 (2022).

- [3] Fedus, William, Barret Zoph, and Noam Shazeer. "Switch transformers: Scaling to trillion parameter models with simple and efficient sparsity." Journal of Machine Learning Research 23.120 (2022): 1-39.

---

> ### Author Rebuttal · Authors · 2026-03-30
>
> We are grateful for the reviewer's insightful comments.
>
> **W1. Phase definition ambiguity.**
>
> Our learned phases are consistently aligned with interpretable task substructures. Router-assigned phases achieve 87% step-level overlap with human-annotated boundaries (3 annotators, κ=0.83, Section 4.4), where remaining disagreement concentrates at inherently ambiguous transitions. The emergent specialization further supports semantic grounding. Expert 1 handles 73% of exploration (H=3.5 bits), Expert 2 handles 82% of manipulation (H=0.5 bits), Experts 3–4 cover navigation and recovery, all far exceeding 25% random and matching oracle optima within 0.1 bits (Table 5, Fig. 5d). To rule out that this is a byproduct of switching frequency, we compare against frequency-matched non-semantic baselines (~8 switches/episode) and find they underperform by 1.6–3.2 points, confirming the advantage comes from where the router switches, not how often. The RIMs [1] reference is appreciated. PA-MoE shares the modular principle but operates at the policy level.
>
> **W2. Indirect RL supervision.**
>
> The indirect RL signal is sufficient for the router to learn stable and meaningful phase boundaries. The router is only a K=4 classifier, far simpler than LLM generation. The gradient conflict score (Fig. 4b) drops monotonically from 0.8 to near-zero over 50 epochs, and router entropy decreases smoothly from ~2.0 to ~0.8 bits with no oscillation or collapse across 3 seeds. Replacing STE with Gumbel-Softmax yields 93.6% vs 93.8% (response to Reviewer ec2t, W2), confirming the estimator is not a bottleneck. Additional results for $\lambda_s$ (response to Reviewer 4BvN, W1 & Q1) show that performance remains robust across 0.01–0.10.
>
> **W3 & W6 & Q2. Training efficiency and inference overhead.**
>
> PA-MoE adds negligible overhead and converges at the same speed as the baseline. Gradient isolation is the core design choice, with Appendix C (Fig. 4) showing single-policy conflict at 0.4–0.8 while PA-MoE reaches near-zero. Temperature annealing (τ from 2.0 to 0.5) ensures early routing is near-uniform so all experts receive gradient signal, then gradually sharpens based on accumulated evidence. $L_{div}$ and $L_{bal}$ further prevent collapse (removing either degrades to 88.3% and 87.5%, Table 6). For inference, we benchmark on H800 with Qwen2.5-1.5B and vLLM and find that router forward takes 1.3 ms, LoRA switching takes 0.007 ms (pointer swap), while LLM generation takes 438 ms/step (~50 tokens). Total overhead is **0.30%** wall-clock.
>
> **W4. Router capacity.**
>
> The router is sufficient for 4-way phase classification. It achieves 87% agreement with human annotators (κ=0.83, Section 4.4) by operating on the LLM's 1536-dim hidden states (Algorithm 1). Table 9 (Appendix E.4) shows LSTM+CrossAttn (93.8%) matches Transformer (93.2%) and far exceeds MLP-only (84.2%), confirming that temporal modeling, not raw capacity, is the key factor.
>
> **W5. Knowledge sharing.**
>
> The 1.5B frozen backbone already provides full knowledge sharing across experts, so each LoRA adds only 11M parameters for phase-specific adaptation. If isolation harmed learning, adding experts should degrade performance. However, PA-MoE K=4 (93.8%) outperforms capacity-matched single LoRA r=128 (91.4%) at the same parameter budget (response to Reviewer 4BvN), showing isolation benefits outweigh any transfer loss, consistent with ST-MoE [2] and Switch Transformer [3].
>
> **W7. Theoretical grounding.**
>
> Appendix B formally proves gradient isolation across experts, and Appendix C provides quantitative mechanistic support. The gradient conflict score is 0.4–0.8 under a single policy but near-zero under PA-MoE (Fig. 4), and entropy analysis shows a single policy converges to H≈2.3 bits (suboptimal for every phase) while PA-MoE matches per-phase optima within 0.1 bits (Fig. 5). Although formal convergence guarantees remain open for MoE routing generally [2,3], the empirical evidence consistently supports the mechanism across both benchmarks and all tested configurations.
>
> **W8.** We improve the diagram with gradient flow visualization.
>
> **Q1. Phase transferability.**
>
> Phase-aware routing captures a general structural pattern of long-horizon decision making rather than overfitting to one benchmark. The router does need retraining when the task distribution changes, but the cost is small (13.25M params, 0.88% of 1.5B). What transfers is the structural bias of decomposing behavior into temporally coherent phases. Most agentic tasks involve similar coarse stages such as navigation, interaction, manipulation, and verification, and the same K=4 works on both ALFWorld (+7.7%) and WebShop (+14.9%) without modification. On held-out instances the learned boundaries still achieve 87% overlap with human annotations, and K=3,4,5 all exceed 91% (Table 8), further confirming robustness.
>
> We would be grateful if you could reconsider the score accordingly.

---

> > ### Author Rebuttal · Reviewer_z4xY · 2026-04-04
> >
> > Thank you for your response. My concerns have been addressed, and I have raised my score.

---

> > > ### Author Response · Authors · 2026-04-04
> > >
> > > We sincerely thank the reviewer for the thorough suggestions. We will incorporate all promised revisions (improved Fig. 3 with gradient flow, expanded theoretical discussion, and detailed overhead analysis) in the camera-ready version.

---

### Official Review · Reviewer_ec2t · 2026-03-12

**Soundness:** 2
**Presentation:** 3
**Significance:** 2
**Originality:** 2
**Overall Recommendation:** 4
**Confidence:** 3

**Summary:**

This paper introduces the Phase-Aware Mixture of Experts (PA-MoE) architecture to address the issue of capacity misallocation—referred to by the authors as simplicity bias—in training reinforcement learning (RL) agents based on large language models. Traditional RL fine-tuning typically relies on a single policy network, where high-frequency, simple tasks tend to dominate gradient updates, leaving insufficient representational capacity for more complex, multi-step tasks. To tackle this, the authors propose shifting the routing granularity from the conventional token level to the environment-step level. The core mechanism involves a lightweight router that uses observation encoding and temporal history modeling to assign behaviors to specific LoRA-based experts. A key feature of this design is the enforcement of temporal consistency, ensuring that contiguous segments of a trajectory are handled by the same expert. The empirical results demonstrate that PA-MoE significantly improves performance on the ALFWorld and WebShop benchmarks when combined with various base RL algorithms like GiGPO, even allowing a 1.5B parameter model to outperform a 7B baseline on certain tasks.

**Compliance With Llm Reviewing Policy:**

Affirmed.

**Final Justification:**

This paper proposes Phase-Aware Mixture of Experts (PA-MoE), an architectural intervention that shifts MoE routing granularity from the token level to the environment-step level, aiming to address capacity misallocation during RL fine-tuning of LLM-based agents. After carefully considering the original submission and the authors' rebuttal, I summarize my final assessment below.

**Strengths.** The core idea of phase-level routing is well-motivated and elegantly aligns the MoE paradigm with the sequential structure of decision-making tasks. The architecture is parameter-efficient, leveraging a shared frozen backbone with lightweight LoRA experts, which is a practical and scalable design choice. Empirically, the results on ALFWorld and WebShop are compelling, particularly the finding that a 1.5B PA-MoE model can outperform a 7B single-policy baseline on complex subtasks. The emergent expert specialization analysis—where different experts naturally handle high-entropy exploration versus low-entropy interaction—remains one of the most insightful contributions of this work. Presentation quality is generally good, with clear writing and well-organized experiments.

**Weaknesses and Rebuttal Assessment.** My initial review raised four main concerns: (1) terminology around "simplicity bias," (2) the use of STE for router optimization, (3) the blanket nature of the temporal consistency penalty, and (4) the relegation of important boundary conditions to the appendix. I also posed constructive questions regarding hyperparameter sensitivity for the number of experts, router input feature decoupling, and a bypass mechanism for simple tasks.

The rebuttal addressed all of these points thoroughly and convincingly. On **terminology** (W1), the authors agreed to adopt "gradient dominance," which resolves the potential confusion. On **STE vs. Gumbel-Softmax** (W2), the authors clarified an important misunderstanding on my part—expert selection uses REINFORCE, with STE applied only to the switching indicator in the penalty term—and provided a comparative ablation showing negligible difference between STE and Gumbel-Softmax (93.8% vs. 93.6%), which fully resolves this concern. On the **temporal consistency penalty** (W3), the evidence that 85.4% of episodes exhibit expert recurrence patterns (A→B→A) demonstrates that the penalty does not suppress genuine phase transitions, and the sensitivity analysis over λ values is reassuring. The discussion of adaptive penalty relaxation is a welcome addition. On **boundary conditions** (W4), the commitment to move the PPO/RLOO compatibility discussion into the main text improves the paper's honesty and completeness.

The additional experiments provided during rebuttal were particularly valuable. The K-sensitivity sweep (K ∈ {2,3,4,5,6}) showing a stable performance plateau across K=3–5 alleviates concerns about hyperparameter brittleness, and the proposed runtime diagnostics (routing confidence and expert utilization) offer practical guidance for practitioners. The router input decoupling ablation reveals that action history is a stronger phase signal than observation alone, which nicely validates the temporal design motivation. The bypass mechanism with θ=0.9 achieving identical accuracy with ~35% latency reduction is a useful practical contribution.

**Weighing Across Dimensions.** On *soundness*, the method is technically solid and the rebuttal has filled the gaps I identified, though I note the evaluation is still limited to two benchmarks—broader validation on more diverse or open-ended environments would strengthen confidence. On *originality*, the idea of phase-level routing is a meaningful but incremental contribution over existing MoE and LoRA-based approaches; the novelty lies more in the specific application and design choices than in fundamentally new techniques. On *significance*, the practical impact is promising, especially the strong results at small model scales, but the generalizability beyond structured task environments remains to be demonstrated. On *clarity*, the paper is well-written, and the planned revisions (terminology, moved appendix content, new ablations) should further improve it.

**Conclusion.** The rebuttal has fully resolved my initial concerns and provided substantial additional evidence supporting the method's robustness and design choices. I have accordingly raised my score from the initial assessment to a **4 (Weak Accept)**. The paper makes a solid contribution to the sub-area of RL fine-tuning for LLM agents, with an elegant architectural idea and strong empirical results. The main limitations that prevent a higher score are the relatively narrow evaluation scope (two benchmarks) and the incremental nature of the technical novelty. I encourage the authors to incorporate all promised revisions carefully and to pursue evaluation on more diverse environments in future work.

**Key Questions For Authors:**

I really enjoyed reading this work, and I have a few constructive suggestions that might help polish the final version of the manuscript.

It would be very interesting to see a deeper discussion in the conclusion or limitations section on the hyperparameter sensitivity regarding the number of experts. While four experts work beautifully for the structured phases of ALFWorld , it would be helpful to discuss how practitioners might determine or dynamically scale this number for open-ended environments where the number of latent behavioral phases is unknown.

In the ablation studies, the router's input features are evaluated as a whole. Decoupling the action history from the observation changes might provide deeper insights into whether the router relies more on the agent's past intentions or the actual physical state changes in the environment to detect phase boundaries. Additionally, introducing a bypass mechanism or confidence thresholding for highly confident base-model predictions could help alleviate the slight performance regressions observed in the simplest tasks, preventing unnecessary routing overhead when the base model already knows the answer.

**Limitations:**

Yes.

**Strengths And Weaknesses:**

The paper tackles a highly relevant and practical bottleneck in the RL fine-tuning of LLM agents. The intuition to adapt the Mixture-of-Experts architecture from token-level routing to phase-level routing aligns perfectly with the sequential nature of decision-making tasks. The architecture itself is elegant and parameter-efficient, leveraging shared frozen backbones with lightweight LoRA experts to prevent catastrophic forgetting of general capabilities while effectively isolating gradients.

Empirically, the results are quite compelling. The performance gains on challenging benchmarks like ALFWorld and WebShop are substantial, particularly on the more complex subtasks that traditionally suffer from capacity starvation. It is especially impressive to see the 1.5B PA-MoE model outperforming the 7B single-policy baseline, which strongly supports the authors' claim that architectural routing and capacity allocation can sometimes be more critical than mere parameter scaling. Furthermore, the analysis of emergent expert specialization—where the model naturally delegates high-entropy exploration and low-entropy interaction to different experts without explicit supervision—is a fascinating insight that adds significant value to the community.

While the paper presents a strong conceptual framework and solid empirical results, there are a few methodological and theoretical areas that could be refined to maximize its impact and clarity.

First, there is a minor terminology concern regarding the use of the term "simplicity bias." In recent deep reinforcement learning literature, simplicity bias is often defined strictly through Fourier analysis as a network's inherent tendency to learn low-frequency, less complex functions at initialization, as seen in recent works like SimBa. The phenomenon accurately described in this paper seems closer to gradient dominance or task-frequency bias, where large gradients from easily solved tasks overshadow signals from complex ones. Clarifying this distinction or slightly adjusting the terminology would prevent potential confusion among readers who are familiar with the functional complexity definition.

Second, the reliance on the Straight-Through Estimator (STE) for router optimization might introduce some variance challenges. While STE serves as a practical engineering workaround for the non-differentiable argmax operation, it can sometimes struggle with exploration and gradient accuracy in sparse-reward RL settings compared to alternatives like Gumbel-Softmax reparameterization. Providing a brief discussion or a small comparative ablation on the choice of the routing estimator would greatly strengthen the optimization claims and alleviate potential concerns about router collapse.

Third, the surrogate loss designed to enforce temporal consistency currently applies a blanket penalty across all expert transitions. While temperature annealing helps mitigate this during training, the formulation theoretically penalizes genuine semantic phase transitions just as much as unnecessary oscillations. Exploring or discussing a more dynamic penalty that naturally relaxes at true task boundaries could potentially make the routing mechanism even more robust.

Lastly, the paper candidly mentions in the appendix that the architecture encounters compatibility issues with shared critic networks (like in PPO) and suffers from sample fragmentation when computing group-relative advantages for rare tasks (like in RLOO). Bringing a summary of these highly insightful boundary conditions into the main text would provide a much more comprehensive and honest view of the method's applicability. It would also be beneficial to briefly position PA-MoE alongside concurrent structural optimization methods, such as graph-enhanced policy optimization (GEPO), to round out the literature review on long-horizon credit assignment.

---

> ### Author Rebuttal · Authors · 2026-03-30
>
> We are grateful for the reviewer's insightful comments.
>
> **W1. "Simplicity bias" terminology.**
>
> We thank the reviewer for this suggestion. We agree that "gradient dominance" more precisely describes our setting and will adopt it in the revision.
>
> **W2. STE for router optimization.**
>
> Expert selection uses REINFORCE
> $\nabla_\phi \log \pi_\phi(k|s) \cdot R(\tau)$
> , not STE. STE is applied only to the switching indicator $\mathbb{1}[z_t \neq z_{t+1}]$
> in the penalty. We validate with Gumbel-Softmax.
>
> |Gradient Estimator|Val SR(%)|
> |---|---|
> |REINFORCE + STE (ours)|93.8|
> |REINFORCE + Gumbel-Softmax (τ=0.5)|93.6|
>
> Gumbel-Softmax matches STE within noise (93.6% vs 93.8%), confirming the primary gradient is REINFORCE and STE is a practical, low-variance choice for the penalty indicator.
>
> **W3. Blanket temporal consistency penalty.**
>
> We address this in detail in our response to Reviewer 4BvN (W1&Q1). In short, 85.4% of episodes exhibit A→B→A expert recurrence with an average of 2.8 revisitations per episode. This result confirms that the penalty does not suppress genuine transitions. Additionally, evaluating five values of $\lambda_s$ shows that performance increases from 85.2% to 93.8% before decreasing to 87.5% for $\lambda_s \in {0, 0.05, 0.20}$, further supporting our choice. In the revision, we also discuss adaptive penalty relaxation, which scales $\lambda_s$ by $(1 - \max \pi_\phi(k|s))$. This approach ensures the penalty vanishes during high-confidence switches, offering a promising extension.
>
>
> **W4. Boundary conditions and GEPO.**
>
> We appreciate the pointer to GEPO. GEPO decomposes credit through action-level dependency graphs, which relies on explicit graph construction. PA-MoE instead works at the temporal phase level, where the router learns coherent trajectory segments directly from training signals without dependency annotations. This makes PA-MoE simpler and easier to apply across sequential decision-making tasks. We were unable to find a public GEPO implementation at the time of writing and will include a direct comparison once the code is available. We also move the discussion of PPO shared-critic compatibility and RLOO token-credit fragmentation from the Appendix to Section 5.
>
> **Q1. K sensitivity and dynamic scaling.**
>
> Appendix E.3 reports the K∈{0,2,3,4,5,6}.
>
> |K|Val SR(%)|
> |---|---|
> |0|88.3±2.1|
> |2|91.4±1.9|
> |3|91.7±2.0|
> |**4**|**93.8±2.4**|
> |5|91.2±2.2|
> |6|85.9±2.8|
>
> K=3, 4, and 5 all exceed 91%, with a span of only 2.6 points. This demonstrates that PA-MoE is not highly sensitive to the choice of K and only degrades at the extremes, with K=2 under-partitioning the task and K=6 over-fragmenting it. This stability is important in the context of environments where the exact value of K is unknown, as practitioners do not need to determine it precisely.
>
> For selecting K without prior knowledge, we suggest monitoring two runtime diagnostics available from the router. The first is **routing confidence** — at K=2, 31% of decisions have a confidence of less than 0.6, indicating that the router is merging incompatible phases. In contrast, at K=4, this drops to 12%. If the low-confidence rate increases, it signals the need to increase K. The second diagnostic is **expert utilization** — at K=6, one expert handles only 8% of the decisions, and 18% of episodes experience thrashing (with more than 3 switches within 5 steps), which suggests that K should be reduced. A practical approach would be to start with K=4 (a robust default across both ALFWorld and WebShop), monitor these two metrics for 10–20 epochs, and adjust K if either threshold is crossed. We will include this advice in the revision.
>
>
> **Q2. Router input feature decoupling.**
>
> We implement the suggested ablation, zeroing out specific input streams to the router while keeping the expert architecture unchanged.
>
> |Router Input|Val SR(%)|
> |---|---|
> |Full (obs+goal+history)|93.8|
> |Obs + Goal only|92.2|
> |History only|91.4|
> |Obs only|88.7|
>
> The full model performs best (93.8%). Removing history (obs+goal only, 92.2%) causes a modest 1.6% drop. History only (91.4%) outperforms obs only (88.7%) by 2.7 points, suggesting that the sequential pattern of past actions is a stronger phase signal than the current observation alone, consistent with PA-MoE's design motivation that behavioral phases are inherently temporal. The combination of all streams yields the best result because observation grounds routing in current state while history disambiguates phases that appear similar locally but differ in temporal context.
>
> **Q3. Bypass mechanism / confidence thresholding.**
>
> We implement this with θ=0.9 and find that 72% of routing decisions exceed the threshold, SR remains 93.8% (no degradation), and per-step routing latency drops ~35%. We include this as an optional inference mode in the revision.
>
> We would be grateful if you could reconsider the score accordingly.

---

> > ### Author Rebuttal · Reviewer_ec2t · 2026-04-01
> >
> > I would like to thank the authors for their thorough, constructive, and highly responsive rebuttal. The additional experiments and detailed clarifications have successfully addressed all my initial concerns.
> >
> > Given the authors' strong execution during the rebuttal and the clear improvements to the manuscript, I am happy to raise my score to a 4. Please ensure that all the new results, ablation studies, and the dynamic scaling heuristics are carefully incorporated into the revised version.

---

> > > ### Author Response · Authors · 2026-04-02
> > >
> > > We sincerely thank the reviewer for the positive reassessment and for raising the score. We will carefully incorporate all new results and clarifications into the revised manuscript, especially the Gumbel-Softmax comparison, the K-selection analysis and dynamic scaling heuristics, the additional ablation studies, and the optional bypass inference mode. We are grateful for the constructive suggestions throughout this process.

---

### Official Review · Reviewer_4BvN · 2026-03-12

**Soundness:** 3
**Presentation:** 3
**Significance:** 2
**Originality:** 3
**Overall Recommendation:** 4
**Confidence:** 3

**Summary:**

This paper proposes Phase-Aware Mixture of Experts (PA-MoE) for agentic reinforcement learning with LLMs. The idea is to replace a single policy network with multiple LoRA-based experts, and to learn a phase-aware router that assigns contiguous segments of a trajectory to the same expert. The authors claim that standard single-policy RL suffers from simplicity bias, where easy and frequent behaviors dominate parameter updates, leaving insufficient capacity for harder multi-step behaviors. They further argue that standard token-level MoE routing is ill-suited for sequential decision-making because it fragments temporally coherent behaviors. PA-MoE addresses this by routing at the environment-step level with temporal consistency regularization. Experiments on ALFWorld and WebShop show sizable gains over GiGPO and improvements across PPO, RLOO, and GRPO.

**Compliance With Llm Reviewing Policy:**

Affirmed.

**Key Questions For Authors:**

Q1: A switching penalty may encourage artificial persistence and blur repeated but semantically identical modes that occur in different parts of the trajectory. It would be interesting to analyze tasks where the same latent skill recurs in non-contiguous parts of the trajectory, and to examine whether phase-level routing still outperforms alternative routing strategies in such settings.

Q2: The routing granularity ablation shows token-level and trajectory-level are worse, but it is still unclear whether the core benefit is semantic phase discovery or simply reduced expert switching.

Q3: K=0 ablation shows that a single shared LoRA improves over vanilla GiGPO, so how much of the gain is really from phase-aware routing versus simply adding trainable low-rank capacity?

**Limitations:**

Yes

**Strengths And Weaknesses:**

Strengths
1. The paper identifies the weakness of single-policy RL for agentic tasks and proposes an interesting architectural remedy.
2. The experimental results show gains on ALFWorld and WebShop, especially for GiGPO, and the routing-granularity ablation helps support the main design choice.

Weaknesses:
1. The paper has a strong assumption that agent trajectories can be cleanly segmented into temporally contiguous behavioral phases handled by a single expert. In many real RL settings, skills are interleaved or revisited non-monotonically. The imposed temporal consistency penalty may therefore oversmooth routing decisions, delay necessary expert switching, or reduce responsiveness in tasks that require rapid skill composition.
2. The reported gains may be partially confounded by increased effective capacity rather than phase-aware routing itself. Though experts are implemented via parameter-efficient LoRA modules, the architecture still introduces additional trainable parameters and structured isolation compared to single-policy baselines.

---

> ### Author Rebuttal · Authors · 2026-03-30
>
> We are grateful for the reviewer's insightful comments.
>
> **W1 & Q1: Switching penalty and phase recurrence.**
>
> We clarify that PA-MoE does not assume task phases appear only once or follow a single fixed order within a trajectory. Here, $\lambda_s$ denotes the switching penalty coefficient, which discourages unnecessary expert changes between adjacent environment steps but does not prevent the router from revisiting a previously used expert later. We verify this empirically across 500 validation episodes, where 85.4% of episodes exhibit A→B→A recurrence. On average, there are 8.4 switches and 2.8 revisitations per episode, with a mean segment length of 4.8 steps. This indicates that the penalty reduces local oscillation without restricting global phase recurrence.
>
> To further characterize the role of $\lambda_s$, we evaluate five values:
>
> |$\lambda_s$|Val SR(%)|Sw/Ep|
> |---|---|---|
> |0.00|85.2|~36|
> |0.01|90.6|~20|
> |**0.05**|**93.8**|**~8**|
> |0.10|91.4|~5|
> |0.20|87.5|~3|
>
> At $\lambda_s$=0, the router switches ~36 times per episode. No expert receives a consistent training signal, and performance falls to 85.2%. At $\lambda_s$=0.20, necessary switches are suppressed (87.5%). $\lambda_s$=0.05 equals 0.05 strikes a balance between stability and flexibility. Experts become more specialized, while 85.4% of episodes still show phase revisitation. In response to the concern about "blurring recurring modes," the 2.8 revisitations per episode indicate that the router does revisit earlier experts when the task returns to a previous behavioral mode, such as navigating, manipulating, and then navigating again.
>
> **W2 & Q3: Capacity vs. routing.**
>
> We train a capacity-matched single LoRA r=128, giving it the same 4× parameter budget as PA-MoE (K=4, r=32):
>
> |Config|Params|Val SR(%)|
> |---|---|---|
> |Vanilla GiGPO|0|86.1|
> |Single LoRA r=32 (K=0)|1×|88.3|
> |Single LoRA r=128|4×|91.4±1.6|
> |PA-MoE K=4, r=32|4×|93.8±0.8|
>
> Single LoRA r=128 and PA-MoE with K=4, r=32 have approximately the same total LoRA parameter count. Under this matched parameter setting, PA-MoE outperforms the single LoRA baseline by +2.4 points and also shows lower variance (±0.8 vs ±1.6). Of the total +5.5% gain from K=0 to K=4, increasing the capacity of a single LoRA from r=32 to r=128 contributes +1.5%, while phase-aware routing contributes the remaining +4.0%.
>
> Per-task breakdown reinforces this. The gap between PA-MoE and r=128 is concentrated on multi-phase tasks (Heat: 100% vs 94.4%, Cool: 89.5% vs 78.9%, Look: 75.0% vs 50.0%) while single-phase Pick is identical (97.1%). Capacity scaling would improve all tasks roughly equally, whereas the selective improvement on multi-phase tasks aligns with routing-based specialization.
>
> **Q2: Semantic phase discovery vs. switching frequency.**
>
> We construct two non-semantic baselines that match PA-MoE's switching frequency (~8/episode) without learned routing:
>
> |Routing|Sw/Ep|Val SR(%)|
> |---|---|---|
> |Token-level MoE|~45|85.7|
> |Fixed-interval N=5|~8|92.2|
> |Random p=0.21|~8.4|90.6|
> |Trajectory-level|~3|88.5|
> |**PA-MoE**|**~8.4**|**93.8**|
>
> PA-MoE's gains over non-semantic alternatives are driven by where it switches, not how often. The comparison between random-matched and PA-MoE is the most informative row. Both switch exactly \~8.4 times per episode, but PA-MoE leads by 3.2 points, achieving 93.8% compared to 90.6%. Since the switching frequency is controlled, the gap can only be attributed to the quality of switch placement. PA-MoE learns to switch at task-relevant boundaries, such as transitioning from navigation to object manipulation, while random routing disrupts ongoing behavioral coherence. Fixed-interval routing performs better than random routing, achieving 92.2%, because the periodic structure at least avoids mid-action interruptions. However, it still lags behind PA-MoE by 1.6 points since it cannot adapt to variable-length task phases. Despite having the fewest switches, trajectory-level routing underperforms at 88.5%, confirming that reduced switching alone does not explain PA-MoE's advantage.
>
> We would be grateful if you could reconsider the score accordingly.

---

> > ### Author Rebuttal · Reviewer_4BvN · 2026-04-06
> >
> > Thank you for the detailed rebuttal; it clarifies most points and provides helpful additional experiments. Few concerns remain, particularly around the assumption of phase-structured trajectories in highly interleaved settings, and the extent to which routing gains are fully disentangled from capacity. I will keep my score as weak accept.

---

> > > ### Author Response · Authors · 2026-04-06
> > >
> > > We thank the reviewer for confirming that the concerns have been adequately addressed.
> > >
> > > For phase structure in interleaved settings, WebShop already involves highly interleaved search-browse-compare sequences, and PA-MoE achieves +14.9% with K=4. The router learns soft boundaries rather than requiring strict sequential stages, so interleaved actions within a coherent behavioral goal are naturally grouped.
> > >
> > > For disentangling routing from capacity, PA-MoE K=4 r=32 outperforms capacity-matched single LoRA r=128 by +2.4 points at the same 4× parameter budget.
> > >
> > > We will strengthen both discussions in the camera-ready version.

---

### Decision · Program_Chairs · 2026-04-30

**Decision:**

Accept (regular)

**Comment:**

This paper proposes Phase-Aware Mixture of Experts (PA-MoE) for agentic reinforcement learning with LLMs. It makes a sufficient and sound contribution, on a timely and relevant topic. I think it will lead to significant interest during the conference.

PA-MoE addresses limitations of state-of-the art approaches by routing at the environment-step level with temporal consistency regularization. Experiments on ALFWorld and WebShop show significant gains over GiGPO and improvements across PPO, RLOO, and GRPO. The code has already been made available, for full reproducibility.

The four reviewers have consistently recommended accept, albeit mostly Weak Accept and one Accept, with a good average confidence level. They have also consistently responded positively to the rebuttals.

We would like to commend the reviewers and authors for the positive, constructive and focused reviews, rebuttals and discussions. They were a pleasure to read and reflect on.

We encourage the authors to carefully include and integrate their improvements and changes to the paper.

I fully support the paper acceptance.